# Single-cell analysis identifies conserved features of immune dysfunction in simulated microgravity and spaceflight

Fei Wu [1,20], Huixun Du[1,2,20], Eliah Overbey[3], JangKeun Kim [3], Priya Makhijani[1,4], Nicolas Martin[1], Chad A. Lerner[1], Khiem Nguyen[1], Jordan Baechle[1], Taylor R. Valentino[1], Matias Fuentealba[1], Juliet M. Bartleson[1], Heather Halaweh[1], Shawn Winer[5,6], Cem Meydan [3], Francine Garrett-Bakelman [7,8], Nazish Sayed [9], Simon Melov [1], Masafumi Muratani [10,11], Akos A. Gerencser[1], Herbert G. Kasler[1], Afshin Beheshti [12,13], Christopher E. Mason [3,14,15,16,21] ✉, David Furman [1,17,18,21] ✉ & Daniel A. Winer [1,2,4,5,19,21] ✉

Microgravity is associated with immunological dysfunction, though the mechanisms are poorly understood. Here, using single-cell analysis of human peripheral blood mononuclear cells (PBMCs) exposed to short term (25 hours) simulated microgravity, we characterize altered genes and pathways at basal and stimulated states with a Toll-like Receptor-7/8 agonist. We validate single-cell analysis by RNA sequencing and super-resolution microscopy, and against data from the Inspiration-4 (I4) mission, JAXA (Cell-Free Epigenome) mission, Twins study, and spleens from mice on the International Space Station. Overall, microgravity alters specific pathways for optimal immunity, including the cytoskeleton, interferon signaling, pyroptosis, temperature-shock, innate inflammation (e.g., Coronavirus pathogenesis pathway and IL-6 signaling), nuclear receptors, and sirtuin signaling. Microgravity directs monocyte inflammatory parameters, and impairs T cell and NK cell functionality. Using machine learning, we identify numerous compounds linking microgravity to immune cell transcription, and demonstrate that the flavonol, quercetin, can reverse most abnormal pathways. These results define immune cell alterations in microgravity, and provide opportunities for countermeasures to maintain normal immunity in space.

Astronauts in low earth orbit (LEO), such as on the international space station (ISS), experience immune dysfunction associated with the microgravity environment. Multiple studies have described immune dysregulation in short or long-term simulated[1–4] or actual microgravity[5–9]. For the most part, such studies have described impaired T-cell responses, coupled with some form of heightened innate immunity[7,10], though some innate immune cells, like natural killer (NK) cells, also show impaired function[11]. Consistent with altered adaptive immunity, potentially due to impaired cytotoxic and Th1 T cell function, and reduced NK cell function, astronauts develop increased reactivation of latent viruses, including herpes viruses (EBV, CMV, VZV)[3,7,12–15]. In one study, viral shedding after 9–14 days of spaceflight was linked to changes in serum cytokines, including a preferential large increase in IL-4 compared to interferon (IFN)γ, indicating a possible shift away from Th1 immunity towards Th2 immunity[16]. Consistently, some astronauts report heightened hypersensitivity reactions, such as increased allergic and Th2-like responses in space[7].

Multiple studies using higher throughput approaches have started to add insight into pathways impacted by spaceflight. In the Twins study[17], a one-year ISS mission altered innate, adaptive, and NK cell-mediated immunity across bulk RNA sequencing analysis. In T cells, increases in DNA methylation were seen in the promoters of *notch3* for CD4[+] T cells, linked to T cell differentiation, and in *scl1a5/asct2*, linked to activation, for CD8[+] T cells. A total of 50 of 62 assayed cytokines were also altered by spaceflight or landing[17]. During a recent multi-omic analysis, including bulk RNA and DNA methylation sequencing, of astronauts and mice in space, mouse organs such as the liver and kidney demonstrated reduced IFN signatures, coupled to altered methylation patterns of these gene sets, while muscles had increased IFNγ, IL-1, and TNF[10]. Serum inflammatory markers from 59 astronauts in this study (and in a similar companion study) showed increased VEGF-1, IGF-1, and IL-1 during spaceflight, which resolved upon returning to Earth[4,10]. This same study also identified mitochondrial dysfunction as a major response of different non-hematolymphoid tissues to spaceflight[10]. More recently, another study using a NASA-developed Rotating Wall Vessel, which was employed in our current work, utilized a 41-parameter mass cytometry approach to show that short-term (18–22 hours) simulated microgravity can dampen NK cell, CD4[+], and CD8[+] T cell responses to Concanavalin A/anti-CD28 stimulation, but potentiates STAT5 signaling to boost Tregs[18].

Despite these important advances, the core fundamental mechanisms, genes, and pathways that are directly altered by microgravity to adversely impact immunity, including at single-cell resolution, are largely unknown. Interestingly, mechanical forces are emerging as critical orchestrators of immune cell function, whereby mechanotransduction tunes immune cell responsiveness to danger signals[19]. Some of these effects occur through environmental modulation of mechanosensing pathways that alter ion currents in cells, metabolism, or directly act on the cytoskeleton[19]. Thus, a spaceflight environment, which alters forces such as gravity, associated hydrostatic pressure, and shear force[20,21] onto immune cells likely directly contributes to immune system dysfunction.

Here, using a common ground-based analog, the NASA developed low shear modeled microgravity Rotating Wall Vessel (RWV)[2,18,22], we examine in depth how short-term (25 hours) exposure to simulated microgravity impacts the human peripheral blood mononuclear immune system in detail at single-cell resolution. Combining this data with validation experiments from mice and crewmembers in LEO, as well as machine learning algorithms, we identify numerous core genes and pathways in immune cells that are altered by simulated microgravity or spaceflight, and identify numerous potential compounds that directly map onto immune cell transcriptional signatures in simulated microgravity.

## Results
### Simulated microgravity alters the transcriptional landscape of individual immune cells
To begin understanding how simulated microgravity impacts immune cell function, we loaded PBMC samples from two young healthy CMV+ donors, one male, and one female, into either RWV simulated microgravity (uG) or normal gravity (1G) static controls for 16 hours of conditioning. The 16-hour-conditioning time point was chosen based on prior work that used approximately the same time and tracked transcriptional or proteomic changes on immune cells to simulated microgravity[1,18]. PBMCs were either left unstimulated or stimulated for an additional 9 hours with R848, a standard TLR7/8 agonist. We chose TLR7/8 as a putative target since it mimics viral infection, and is expressed on most human immune cells, including T cells[23]. Using this methodology, we next developed a single-cell atlas of 55,648 human PBMCs exposed to these conditions.

In the unstimulated state, after 25 hours of simulated microgravity, we identified 28 clusters of immune cells visualized by UMAP

(Uniform Manifold Approximation and Projection), including cell types such as mucosal associated invariant T cells (MAIT cells), double negative T cells, γδ T cells, innate lymphoid cells, and plasmacytoid dendritic cells, which have rarely been studied in simulated microgravity (Fig. 1A). Simulated microgravity altered proportions of immune cell clusters to a mild extent, with B intermediate cells, and MAIT cell proportions being most negatively impacted, and CD14[+] monocytes, and CD4[+] T effector memory (TEM) cells being most increased based on percent change (Fig. 1B). Across all immune populations, simulated microgravity altered expression of over 4500 genes with adj *P* cutoff of <0.05 (Supplementary Data 1). This list was refined to a core list of ~375 differentially expressed genes (DEGs) with an additional cutoff of |log2FC|> 0.1. This list was further condensed to visualize on a Volcano plot with |log2FC|> 0.25 (Fig. 1C), showing only the very top positively and negatively altered genes. Volcano plots of DEGs for individual immune cell clusters are shown in Supplementary Fig. 1. Across all immune cells, some of the most induced genes in simulated microgravity included acute response genes such as *s100a8*, *s100a9*, *s100a12*, *thbs1*, heat-shock genes such *hsp90ab1*, chemokines like *ccl2*, *ccl4*, iron storage genes (*fth1*, *ftl*), and matrix metalloproteinases (*mmp9*). The most reduced genes in simulated microgravity included interferon response (*stat1*) and associated guanylate binding proteins (*gbp1*), and cold shock genes (*rbm3*, *cirbp*). Expression of the top DEGs (with mitochondrial encoded genes excluded for visual simplicity) across 22 populations of immune cells are shown in Fig. 1D. CD14[+] classical monocytes, CD16[+] nonclassical monocytes, and natural killer (NK) cells exhibited the most pronounced changes across major gene sets, consistent with short term simulated microgravity's direct effect at reprogramming transcriptional changes most prominently in innate immunity. Consistently, using single-cell trajectory analysis, we identify numerous trajectories mainly in the innate immune cell clusters, especially the monocyte cluster, in response to simulated microgravity. Trajectory analysis is used to construct a path that describes how cells move through different states, and the numerous states seen in the monocyte cluster in simulated microgravity may reflect an increased capacity to generate distinct transcriptional states to simulated microgravity (Fig. 1E).

Ingenuity pathway analysis (IPA) (Fig. 1F, Supplementary Data 2) generated using our core list of 375 genes from the overall populations, as well as the DEGs in major immune cell types (Supplementary Data 3) revealed that monocytes, conventional dendritic cells type 2 (cDC2)s, double negative (dn)T cells and NK cells show the most notable pathway alterations. Major pathways altered by simulated microgravity across immune cells included reductions in oxidative phosphorylation, interferon signaling like protein kinase R (PKR) in interferon response, nuclear receptor signaling (LXR/RXR, PPAR, AHR), RHOA and pyroptosis signaling, as well as increases in BAG2 (heat-shock protein 70 interactor) signaling, fibrosis signaling, actin-based motility, RAC, HIF1 signaling, acute phase response, oxidative stress and sirtuin signaling, amongst others.

Given that multiple pathways we detected were associated with inflammatory processes linked to aging (i.e., increased innate immunity coupled to reduced adaptive immunity), we next determined whether acute exposure to simulated microgravity mimicked inflammatory aging processes in immune cells. We mapped the gene expression signatures of individual immune cells, and overall immune signatures, against two recently developed inflammatory signatures of aging, the inflammatory age (iAge) clock[24], and the SenMayo list of senescence associated secretory inflammatory products[25]. Simulated microgravity induced a significant enrichment in inflammaging related genes, consistent with the notion that short term simulated microgravity can induce aging-like inflammatory changes in unstimulated immune cells (Fig. 2A, B, Supplementary Fig. 2A). Next, because both spaceflight and aging are associated with reactivation of latent viruses, we mined the meta-transcriptome of our single-cell analysis with meta-

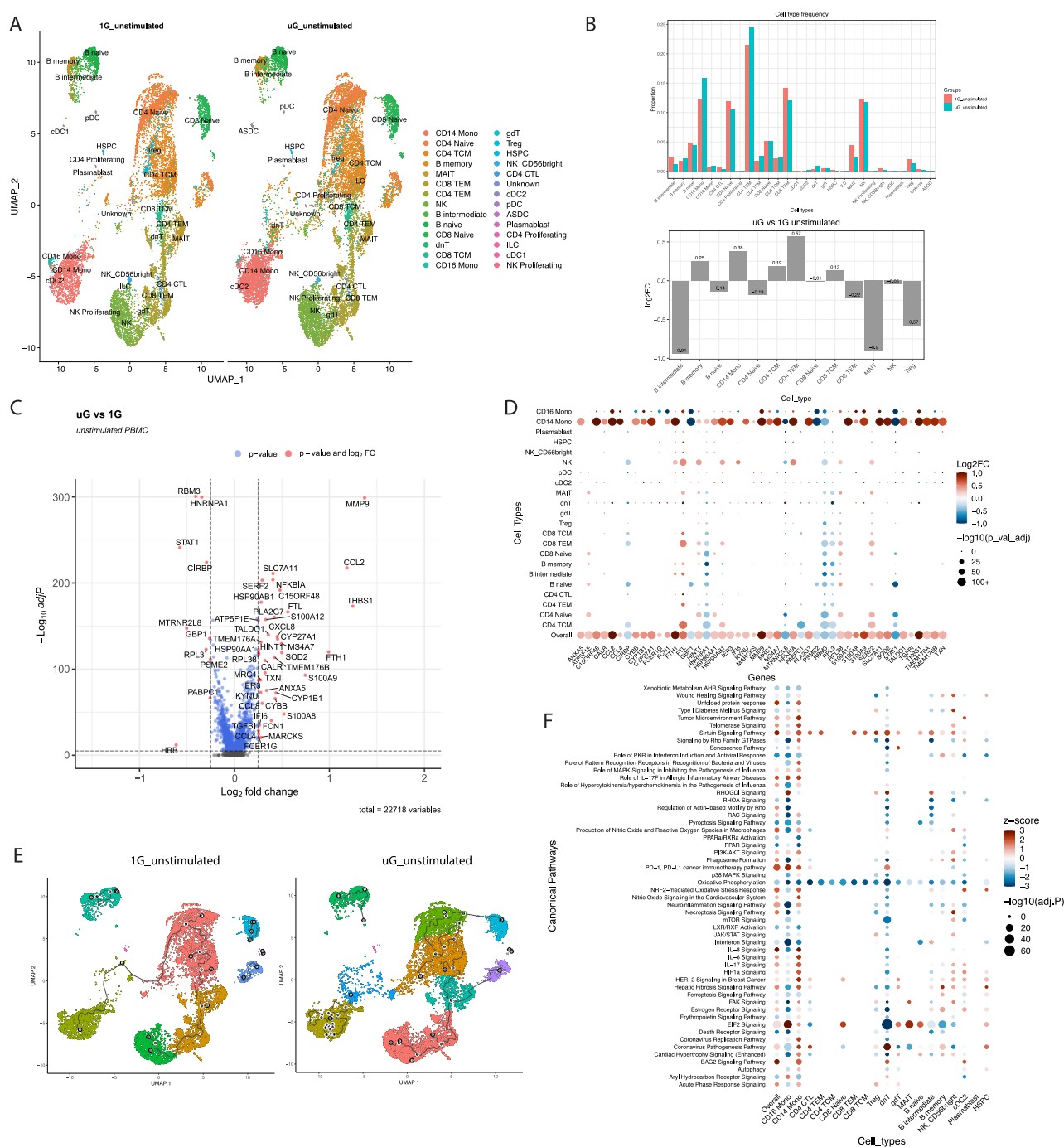

**Fig. 1 | Simulated microgravity without pattern recognition receptor stimulation alters the single-cell transcriptional landscape of human PBMCs. A** UMAP plot of unstimulated PBMCs single-cell transcriptomes (10X Genomics), pooled together from a male (36 years old) and a female (25 years old) donor, that underwent either 1G or simulated microgravity (uG) for 25 hours total. Cells were resolved into 28 distinct clusters. **B** Quantification of relative abundance of each cluster of single PBMCs by percentage, or log2Fold Change (FC) between simulated uG and 1G conditions. Source data are provided with this paper. **C** Volcano plot of differentially expressed genes (DEGs) across all immune cell types between uG and 1G; DEGs (including log2FC and adj. *p*) were calculated by the MAST method. The Benjamini–Hochberg (B-H) method was used for multiple comparison adjustments. Adjusted *p* value (adj. *p*) cutoff is 0.05, and log2FC cutoff is 0.25. **D** Dot plot showing the top DEGs from **C** and their expression levels across 22 immune cell

populations. DEGs (including log2FC and adj. *p*) were calculated by the MAST method. *P* values were adjusted by the B-H method. Spot color reflects Log2FC of uG vs 1G, while spot size shows −log10 (adj. *p*). **E** UMAP of trajectory analysis of 1G and simulated uG unstimulated PBMCs. White circles represent the root nodes of the trajectory. Black circles indicate branch nodes, where cells can travel to a variety of outcomes. Light gray circles designate different trajectory outcomes. **F** Canonical pathway enrichment analysis obtained from Ingenuity Pathway Analysis (IPA) is shown across 19 immune cell clusters. Spot color reflects IPA z-score enrichment of simulated uG vs 1G, with red meaning predicted activation of the pathway in uG and blue meaning repression of the pathway in uG. Spot size shows the level of significance via −log10 (adj. *p*). Adj. *p* was calculated by the Fisher's Exact Test (right-tailed) followed by B-H adjustment.

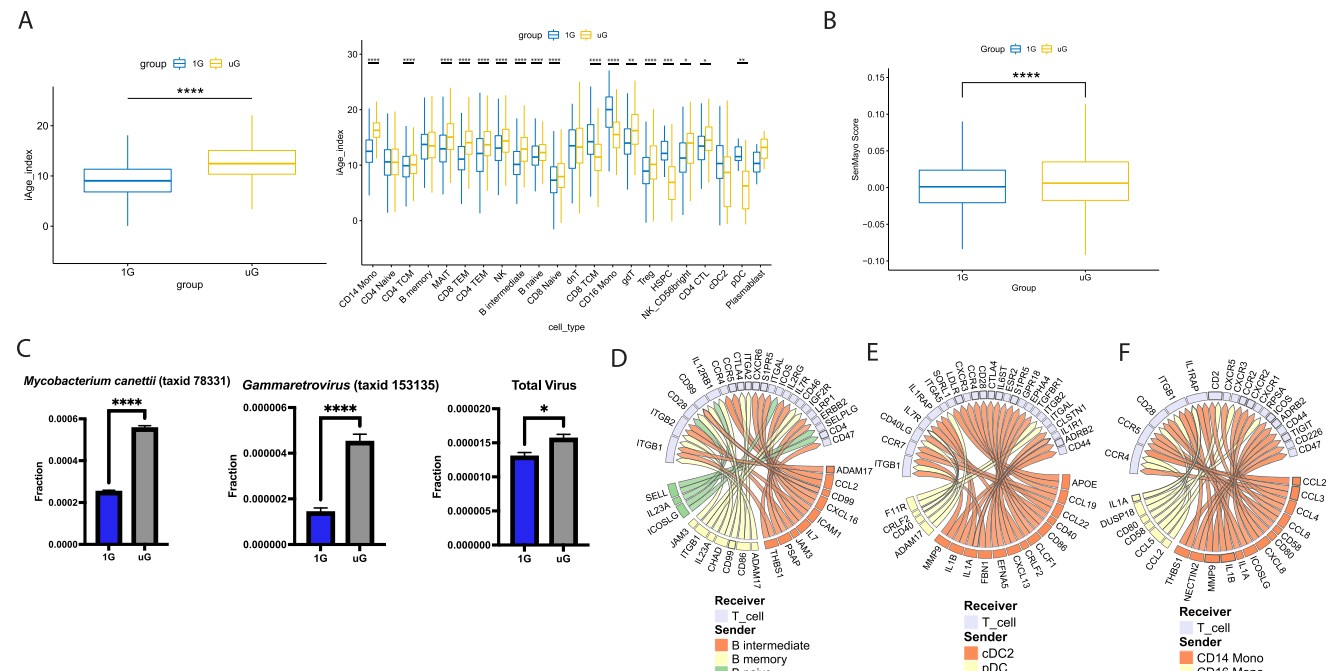

**Fig. 2 | Simulated microgravity induces predictive functional alterations in immune cells. A** Differences in iAge index between all cell types (left) and across 22 individual immune cell types (right) at 1G or simulated uG. Each box spans from the 25th to 75th percentiles (interquartile range, IQR), and features a median value denoted by a horizontal line. The whiskers extend to values within 1.5 times the IQR range from the 25th and 75th percentiles. 1G group ($n = 11,934$ cells examined over 2 independent experiments) is shown in blue and uG group ($n = 16,568$ cells examined over 2 independent experiments) is shown in yellow. The median for 1G group is 9.1, with min = −5.8 and max = 22.4. The median for uG group is 12.5, with min = −1.1 and max = 26.2. Two-tailed Mann–Whitney test ($p$ value < 2.2e-16). ****$p \leq 0.0001$, ***$p \leq 0.001$, **$p \leq 0.01$, *$p \leq 0.05$. Source data are provided with this paper. **B** Differences in cellular senescence secretory product score, calculated from the SenMayo gene set, between all cell types at 1G or simulated uG. 1G group's ($n = 11,934$ cells examined over 2 independent experiments) median is 0.001, with min = −0.084 and max = 0.302; uG group's ($n = 16,568$ cells examined over 2 independent experiments) median is 0.006, with min = −0.091 and max = 0.569. Two-tailed Mann–Whitney test ($p$ value < 2.2e-16). ****$p \leq 0.0001$. Source data are provided with this paper. **C** Meta-transcriptome detection of mycobacteria, retrovirus, and total virus abundance in 1G ($n = 11,934$ cells examined over 2 independent experiments) and uG ($n = 16,568$ cells examined over 2 independent experiments) conditions. The bar plot shows the mean with an error bar representing the standard error of the mean (SEM). For *Mycobacterium canettii* ($p$ value < 0.0001), mean of 1G = 2.5e-4 ± 9.0e-6 and uG = 5.5e-4 ± 1.4e-5. For *Gammaretrovirus* ($p$ value < 0.0001), mean of 1G = 1.4e-6 ± 2.0e-7 and uG = 4.5e-6 ± 3.4e-7. For the total virus ($p$ value = 0.0421), mean of 1G = 1.3e-5 ± 6.3e-7 and uG = 1.6e-5 ± 6.6e-7. Two-tailed Mann–Whitney test; *$p \leq 0.05$, ****$p \leq 0.0001$. Source data are provided with this paper. (D to F) NicheNet predicted significant ligand-receptor interaction between total T cells (Receiver) and the antigen-presenting cells (Sender) as **D** B cells, **E** DCs, and **F** monocytes in uG vs 1G condition (i.e., induced in uG over 1G).

transcriptome detector (MTD) pipeline[26]. Surprisingly, we saw that as little as 25 hours of simulated microgravity could induce the transcription of latent retroviruses and mycobacteria within human immune cells (Fig. 2C, Supplementary Figs. 2B, C and 3), directly implicating microgravity itself as a contributing trigger for latent pathogen activation. We confirmed the meta-transcriptome results with a different alignment tool, and we could still detect increases in *Gammaretrovirus* and *Mycobacterium canettii* transcripts seen with MTD pipeline (Supplementary Fig. 2B, C).

Finally, as we identified strong changes in gene expression pathways linked to innate cells, including those with the capacity to present antigen, we leveraged this knowledge to utilize NicheNet[27] algorithms to generate a comprehensive predicted ligand:receptor interactome map of human antigen-presenting cells (APCs, plus plasmacytoid dendritic cells) and T cells in simulated microgravity vs 1G (Fig. 2D–F). Across APC donors and recipient T cells, we identified numerous significantly predicted ligand-receptor interactions to be elevated in simulated microgravity vs 1G. For instance, monocytes and dendritic cells induced IL-1 proteins while some B cells provided IL-23A, and IL-7. All APCs provided unique chemokine signals to T cells. *Mmp9*, *ccl2*, and *thbs1* were amongst our most significantly induced genes in simulated microgravity, and the products of these genes show differential predicted receptor expression (e.g., CD44, CD47, ITGB1, CCR4, CCR5) in T cells (Supplementary Figs. 4–6) but all show predicted enhanced target gene expression in T cells. Thus, while simulated microgravity

itself likely induces direct transcriptional changes in immune cells, we cannot exclude local paracrine effects of secreted products from one immune cell to another also contributing to our overall gene expression and pathway changes.

After stimulating PBMCs with a TLR7/8 agonist in 1G and simulated microgravity, we characterized 23 clusters of immune cells by UMAP (Fig. 3A). In contrast to the unstimulated conditions, stimulation with a TLR7/8 agonist induced a robust preferential expansion of CD4$^+$ central memory (TCM) cells (Fig. 3B). The microgravity itself impacted differential response to stimulation. Consistent with previous reports, simulated microgravity dampened expansion/responses of NK cells, and CD8$^+$ TEM cells to a lesser extent in the donors examined[18], as well as MAIT cell numbers, a cell type with previously unknown responses to microgravity. Simulated microgravity drove a preferential increase in CD14$^+$ monocytes over 1G controls, indicating that this cell type is especially sensitive to the combination of simulated microgravity and TLR7/8 activation.

Across all cell types, the combination of simulated microgravity and TLR7/8 stimulation altered the expression of over 9000 differentially expressed genes (DEGs) with adj $P$ cutoff of <0.05 (Supplementary Data 4). As with the unstimulated data, we refined this list to a core gene list of ~317 DEGs based on |log2FC| of >0.2. This list was further reduced to visualize on a Volcano plot with |log2FC|>0.25 (Fig. 3C), showing only the most positively and negatively altered genes. Some of the most induced genes by

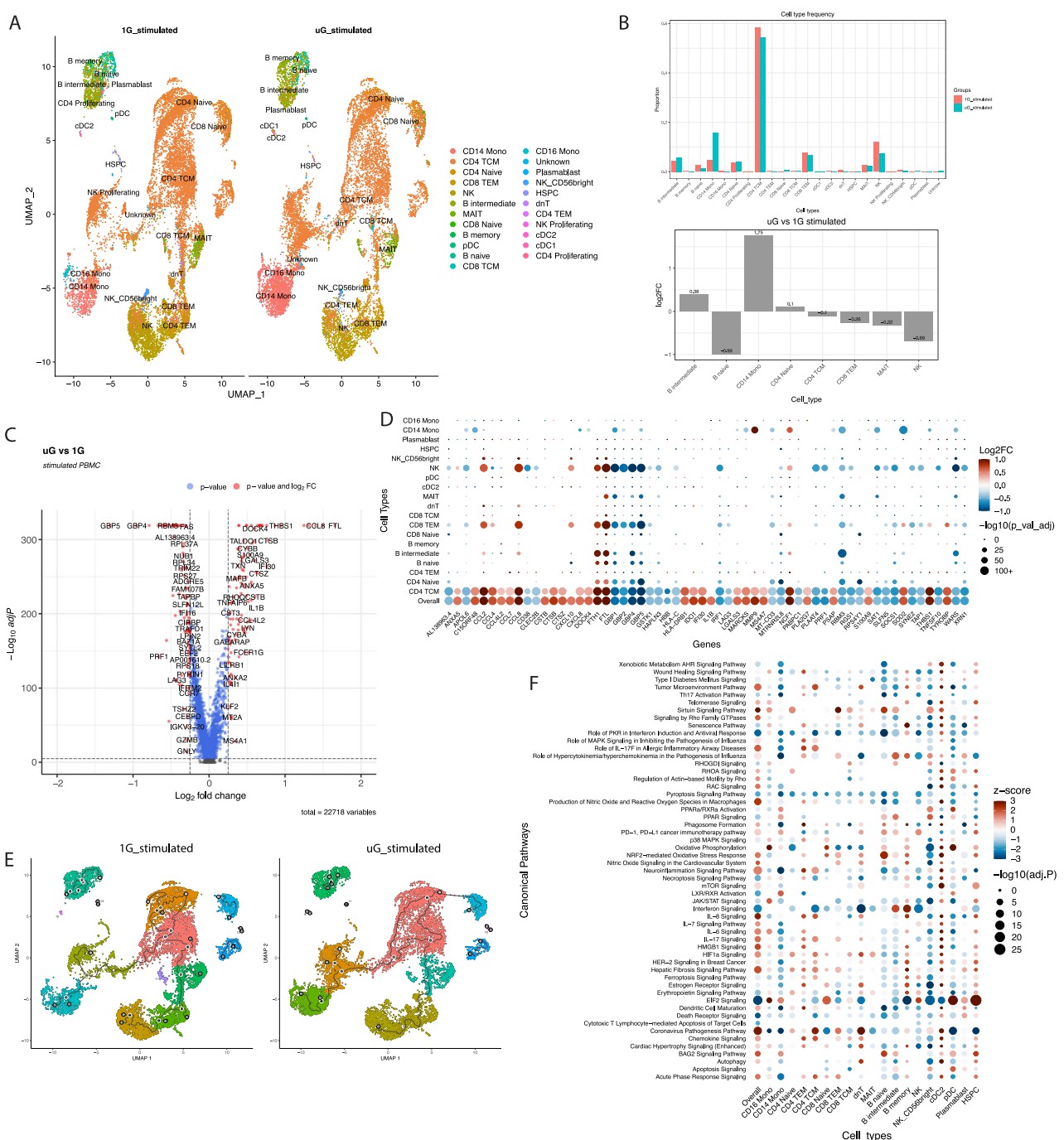

**Fig. 3 | Simulated microgravity induces a distinct single-cell transcriptional landscape of human PBMCs following TLR7/8 stimulation. A** UMAP plot of TLR7/8 agonist stimulated (9 hours stimulation + 16 hours conditioning prior to stimulation = 25 hours total culture) PBMCs single-cell transcriptomics, pooled from a male (36 years old) and a female (25 years old) donor, that underwent either 1G or simulated uG. Cells were resolved into 23 distinct clusters. **B** Quantification of relative abundance of each cluster of single PBMCs by percentage, or log2FC between stimulated uG and 1G conditions. Source data are provided with this paper. **C** Volcano plot of DEGs across all immune cell types between TLR7/8 agonist simulated uG and 1G; DEGs (including log2FC and adj. *p*) were calculated by the MAST method. *P* values were adjusted by the B-H correction. Adj. *p* cutoff is 0.05, and log2FC cutoff is 0.25. **D** Dot plot showing the top DEGs from **C** and their

expression levels across 19 immune cell populations. DEGs (including log2FC and adj. *p*) were calculated by the MAST method; *p* values were adjusted by the B-H correction. Spot color reflects log2FC of TLR7/8 agonist simulated uG vs 1G, while spot size shows −log10(adj. *p*). **E** UMAP of trajectory analysis of 1G and simulated uG TLR7/8 agonist stimulated PBMCs. Filled circle nomenclature was described in Fig. 1E. **F** Canonical pathway enrichment analysis obtained from IPA is shown across 19 immune cell clusters. Spot color reflects IPA z-score enrichment of TLR7/8 agonist-activated simulated uG vs 1G, with red meaning predicted activation of the pathway in simulated uG and blue meaning repression of the pathway in simulated uG. Spot size shows the level of significance via −log10 (adj. *p*). Adj. *p* was calculated by the Fisher's Exact Test (right-tailed) followed by B-H adjustment.

simulated microgravity over 1G in response to TLR7/8 agonist included cytokines and chemokines, such as *ccl8*, *ccl4*, *ccl7*, *cxcl8*, and *il1b*, and acute response proteins like *s100a8*, *s100a9*, *s100a11*, and *thbs1*. Additional genes induced in simulated microgravity were linked to tryptophan breakdown (*ido1*), mitochondrial antioxidant defense (*sod2*), the cytoskeleton (*rhoq*), and iron storage genes like *fth1*, *ftl*. The most downregulated genes when comparing simulated microgravity to 1G during TLR7/8 stimulation, included genes belonging to guanylate binding proteins (*gbp1*, *gbp2*, *gbp4*, *gbp5*), which were the most reduced set of genes by fold change and adj P, as well as interferon pathway genes, like *irf1*, *stat1*, *isg20*, *ifi16*, cold shock genes (*rbm3*, *cirbp*), cell killing genes (*prf1*, *gzmb*) and T/NK cell activation markers like *cd69*. Many of these genes were consistently altered by simulated microgravity alone without stimulation, indicating a conserved response, even in the setting of additional exogenous stimulation with a TLR ligand. Expression of top DEGs across 19 populations of immune cells is shown in Fig. 3D. Volcano plots of DEGs for individual immune cell clusters are shown in Supplementary Fig. 7. CD14+ monocytes, NK cells, CD8+ TEM, and CD4+ TCM cells showed the most significant changes in the top most altered genes induced by TLR7/8 agonist stimulation in simulated microgravity. Interestingly, using single-cell trajectory analysis (Fig. 3E), we identified fewer trajectories in simulated microgravity stimulated with TLR7/8 compared to the 1G control. These findings suggest that under simulated microgravity, cells display reduced differentiation states in response to stimulation.

IPA results (Fig. 3F, Supplementary Data 5) generated using our core list of approximately 317 genes from the overall populations, as well as the DEGs in major immune cell types (Supplementary Data 6) demonstrated that nearly all immune cells show changes across numerous pathways during microgravity and TLR7/8 induction. Major pathways reduced across most immune cells in simulated microgravity included PKR in interferon response (and associated eif2 signaling), interferon signaling, JAK/STAT signaling, pyroptosis signaling, cytotoxic T cell mediated killing of target cells and death receptor signaling. Major pathways induced by short term simulated microgravity included sirtuin signaling, fibrosis signaling, signaling by Rho GTPases, BAG2 (heat-shock protein 70 interactor) signaling, HIF1α signaling, acute phase response and associated HMGB1 signaling, amongst others. These pathways are consistent with microgravity facilitating innate like inflammation at the expense of interferon driven adaptive immunity and adaptive immune effector function (e.g., CD8+ T cell killing). Despite some similarities in pathways altered to simulated microgravity alone (Fig. 1F), we actually detected a lower iAge score globally across all immune populations in simulated microgravity plus TLR7/8 compared to 1G controls (Fig. 4A). While the reason for this finding is unclear, it appears to have been driven by a highly significant reduction in score by naive B cells, naive CD4+ T cells, and reductions in lesser studied PBMC populations, hematopoietic stem and progenitor cells (HSPC)s and double negative T cells. Lower iAge also could be reflective of altered immune activation in simulated microgravity, such as seen in CD16+ monocytes (Fig. 4A, right panel). Despite a reduction

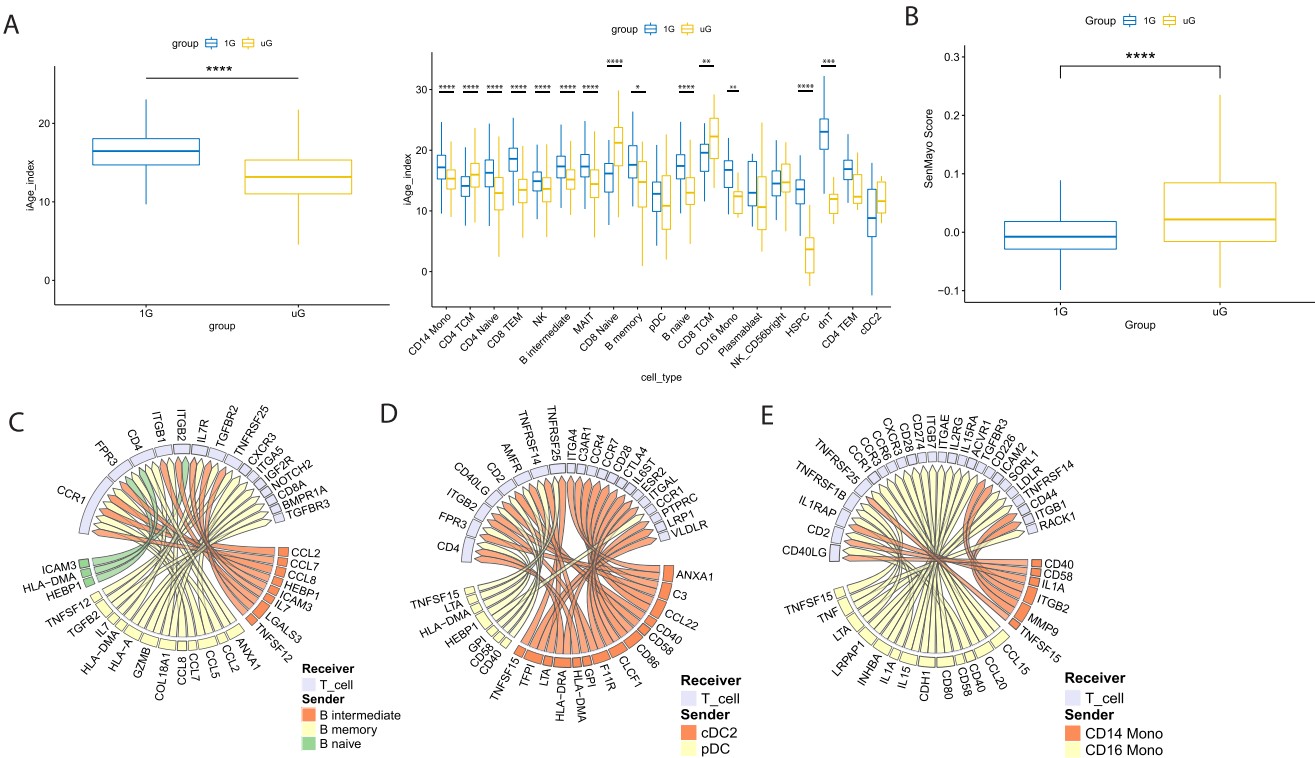

**Fig. 4 | Simulated microgravity induces predictive functional alterations in immune cells following TLR7/8 stimulation. A** Differences in iAge index between all cell types (left) and across 19 individual immune cell types (right) after TLR7/8 agonist activation in 1G or simulated uG. Data are presented in the same way as described in Fig. 2A. 1G group (*n* = 14,916 cells examined over 2 independent experiments) is shown in blue and uG group (*n* = 12,230 cells examined over 2 independent experiments) is shown in yellow. The median for 1G group is 16.4, with min = 2.9 and max = 25.9. The median for uG is 13.2, with min = −1.7 and max = 23.9. Two-tailed Mann–Whitney test (*p* value < 2.2e-16). ****p ≤ 0.0001, ***p ≤ 0.001, **p ≤ 0.01, *p ≤ 0.05. Source data are provided with this paper. **B** Differences in

cellular senescence secretory product score, calculated from the SenMayo gene set, between all cell types with TLR7/8 agonist activated 1G or simulated uG. 1G group's (*n* = 14,916 cells examined over 2 independent experiments) median is −0.008, with min = −0.099 and max = 0.588; uG group's (*n* = 12,230 cells examined over 2 independent experiments) median is 0.022, with min = −0.095 and max = 0.551. Two-tailed Mann–Whitney test (*p* value < 2.2e-16), ****p ≤ 0.0001. Source data are provided with this paper. **C–E** NicheNet predicted significant ligand-receptor interaction between total T cells (Receiver) and the antigen-presenting cells (Sender) as **C** B cells, **D** DCs, and **E** monocytes in TLR7/8 agonist activated simulated uG vs 1G condition (i.e., induced in uG over 1G).

in iAge, we still observed an increased SenMayo score in simulated PBMCs (Fig. 4B, Supplementary Fig. 8), illustrating different compositions of genes in these two gene sets.

NicheNet analysis across major APC types to T cells post TLR7/8 agonist in simulated microgravity vs 1G (Fig. 4C–E, and Supplementary Figs. 9–11) illustrated some of the significant cytokines, chemokines, surface molecule ligands and receptors used in simulated microgravity upon TLR7/8 stimulation. Compared to the unstimulated interactome (Fig. 2D–F), we saw increased production and diversity of inflammatory cytokines and chemokines used. We again noted IL-1 produced in APCs, but also noted increased TNF superfamily products like TNF, TNFSF12 (TNF-related weak inducer of apoptosis, TWEAK) and TNFSF15 (vascular endothelial growth inhibitor, VEGI), and lymphotoxin (LTA) preferentially produced to modulate T cell function.

Next, we assessed the differential responsiveness of immune cells to TLR7/8 stimulation (Supplementary Fig. 12 and Supplementary Data 7). Under 1G conditions, stimulation led to a marked induction of CD4$^+$ TCM, at the expense of CD14$^+$ monocytes and CD4$^+$ naive T cells, coupled to an expected pronounced inflammatory gene signature, including marked induction of interferon inducible genes, *gbp* transcripts, and chemokines across most immune cell populations (Supplementary Fig. 12A–E). Under simulated microgravity, TLR7/8 stimulation also induced CD4$^+$ TCM, at the expense of naive CD4$^+$ T cells, though proportions of CD14$^+$ monocyte populations did not reduce as seen in 1G (Supplementary Fig. 12F, G). In simulated microgravity, TLR7/8 stimulation also induced a robust expression of inflammatory genes, including interferon inducing genes across most cell types (Supplementary Fig. 12H–J). Next, to determine the sensitivity of individual immune cell populations to TLR7/8 agonist in 1G vs simulated microgravity, we compared the differences in responsiveness to stimulation. We subtracted the fold change induction in 1G from induction in simulated microgravity to determine sensitivity to stimulation. Remarkably, across overall immune cells, we see a pattern of reduced responsiveness to TLR7/8 simulation in simulated microgravity to most of the highest genes induced at 1G (Supplementary Fig. 13A), with T cells and NK cells showing the most reduced inflammatory gene induction. T cells, NK cells, and overall across all cells exhibited blunting of induction of numerous genes in interferon signaling, and *gbp* genes in simulated microgravity in response to TLR7/8 agonist. Interestingly, monocytes tended to maintain such responses better in simulated microgravity, consistent with their predisposition to some inflammatory pathways in simulated microgravity. Some chemokines, such as *ccl3, ccl4, ccl8*, and *cxcl10* appeared to be induced better in simulated uG across overall immune cell populations, though monocytes actually showed reduced induction of some of these chemokines, likely due to their capacity to produce them in simulated microgravity without stimulation (Fig. 1D). Nonetheless, the overall effects in sensitivity to stimulation in the "overall" category of immune cells largely followed the same pattern seen in the total magnitude of response of stimulated microgravity vs stimulated 1G (Fig. 3D).

Next, we sought to identify genes uniquely altered by simulated microgravity irrespective of stimulation, as well as genes unique to microgravity under stimulation. First, to identify unique genes, regardless of stimulation, altered by simulated microgravity, we plotted out the most significant DEGs by the absolute sum of fold change under both stimulated and unstimulated conditions (Supplementary Fig. 13B, Supplementary Data 8). Across overall immune cells, regardless of stimulation, we still identified conserved increases in chemokines and acute response factors, coupled to reduced *gbp* expression and other interferon genes (e.g., *irf1, stat1*) imparted by simulated microgravity. Next, we mapped out the overall overlap of gene signatures and common genes between post-stimulation in 1G vs simulated microgravity (Supplementary Fig. 13C–E and Supplementary Data 9), and found that microgravity imparts a number of unique genes to TLR ligation that are not seen in 1G. Overall, these findings

identify core-conserved DEGs specifically sensitive to simulated microgravity as well as unique signatures to simulated microgravity.

Finally, we assessed if sex plays a role in the magnitude of response to simulated microgravity. In the unstimulated state, female cells showed only slightly more DEGs induced, while the male cells had slightly more genes reduced (Supplementary Fig. 14A). Male NK cells and monocytes were more sensitive to microgravity while female B cells showed more sensitivity than male B cells. Upon stimulation, male cells overall were more sensitive to simulated microgravity, especially in having more downregulated DEGs (Supplementary Fig. 14B). Volcano plots of DEGs across all cell types between the female and male are shown in Supplementary Fig. 14C–F and Supplementary Data 10. In both male and female cells, acute phase response and inflammatory genes like *mmp9, ccl2, s100a8*, and *thbs1* were among the most induced genes, while reduced interferon regulators like *stat1* and reduced cold shock genes like *rbm3* and *cirbp* were consistently downregulated in both sexes in simulated microgravity. Upon stimulation, both sexes again show increases in the total magnitude of acute inflammatory, reactive oxygen species (ROS)-related, and acute phase genes like chemokines, *thbs1, mmp9, ncf1,* and *sod2* in simulated microgravity coupled to reduced interferon, *gbp*s, cold shock, and some ribosomal protein genes in simulated microgravity. Many of these changes are reflected in IPA pathway analysis by sex (Supplementary Fig. 14G) and many of these core features were also conserved when data from sexes were pooled (Figs. 1F, 3F).

## Single-cell validation identifies core features of immune dysfunction in microgravity and spaceflight

Next, to better validate conserved genes and pathways from our single-cell signatures without TLR7/8 agonist in simulated microgravity, we compared our core signature of 375 DEGs against additional datasets. First, we repeated experiments in a validation cohort of freshly isolated PBMCs from young donors (n = 6, age range 20–46), and spun these PBMCs for 25 hours, prior to performing bulk RNA-seq analysis. Using CIBERSORTx[28], we first mapped predicted changes in population frequency and were able to confirm increased CD14$^+$ monocyte frequencies in simulated microgravity, consistent with our single-cell data (Supplementary Fig. 15, Supplementary Data 11). Between validation samples spun in simulated microgravity vs 1G controls, we identified 2149 genes differentially expressed (Fig. 5A, Supplementary Data 12). Despite the variability of data inherent to bulk RNA-seq of different populations of cells between donors, we still saw a highly significant correlation in normalized gene counts per specific gene between datasets both at 1G and in simulated microgravity (Fig. 5B). Moreover, we identified overlap in over 28% of our core signature genes (same directionality) across all immune cells from our single-cell analysis (106/375 = 28.3%) (Fig. 5C). Many of the overlapping genes induced were consistent with our most robustly altered core pathways from the single-cell data. For instance, we saw shared overlapping genes induced in acute immune responses (such as *s100a8, s100a12, thbs1, il1b*), chemokines (like *cxcl8*), heat-shock proteins (*hsp90aa1, hspa1a, hspb1*), autophagy (*atg7*) and the actin cytoskeleton (*rhou*). Overlapping reduced genes in simulated microgravity, like in our single-cell datasets, included interferon response (*stat1, irf1*) and associated guanylate binding proteins (*gbp1, gbp2, gbp4, gbp5*), and cold shock genes (*rbm3*), amongst others. Overall, there was a highly significant enrichment and over representation of our core single-cell DEGs across our bulk validation cohort by Fisher's Exact Test for gene overlap (Fig. 5C).

We next sought to validate overlapping genes against mice and people flown in LEO. While multiple stressors exist in LEO, the proximity to Earth and the presence of the Earth's magnetic field negates some effects of galactic cosmic rays, especially at the altitude of the ISS. Thus, microgravity plays an important role in driving phenotypic changes in LEO. To accomplish this goal, we first mined data from

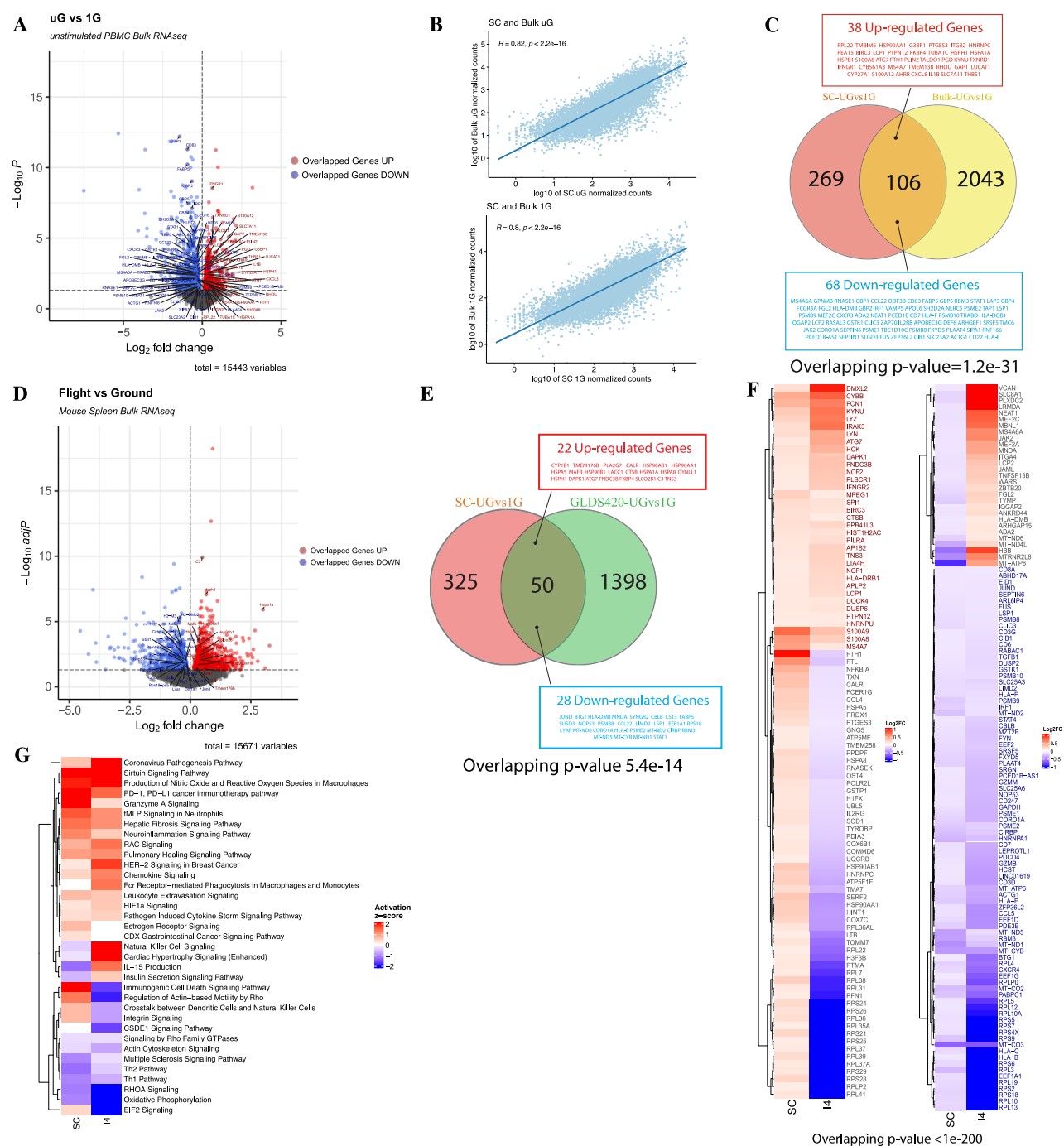

NASA's GeneLab database for its largest study looking at a major immune organ, the spleen, in mice flown on the ISS. The GLDS-420 study provides data from the spleens of ten mice housed on the ISS for 33 days compared to ten ground controls. Though this cohort represents longer exposure to microgravity than our single-cell data's more acute exposure, any overlapping genes could represent persistent microgravity-sensitive immune cell genes across longer duration exposure. From the GLDS-420 dataset, we identified 1448 significant DEGs (Fig. 5D), of which 50/375 (13.3%) overlapped in the same direction as our single-cell core list (Fig. 5E). Interestingly, many of the overlapping genes were represented as part of altered core pathways from the single-cell data. For instance, we saw shared induced overlapping genes in acute immune responses or complement (such as *c3*), autophagy (*atg7*), heat-shock responses (*hsp90ab1*, *hsp90aa1*, *hspa1a*, *hspa1*), and the cytoskeleton (*dynll1*). Overlapping reduced genes

included interferon response (*stat1*), and again, cold shock genes (*cirbp*, *rbm3*), amongst others. Overall, we saw a significant enrichment in our core single-cell DEGs across mouse spleens flown in space by Fisher's Exact Test for gene overlap (Fig. 5E). Pathway analysis with IPA was next performed to identify major canonical pathways altered across all four complete datasets (single-cell unstimulated, single-cell stimulated with TLR7/8 agonist, Bulk RNA-seq validation unstimulated, and GLDS-420), including overlapping pathways shared across all datasets. These pathways are displayed in a heat map for comparison (see below), and will be described at that point.

To better translate the usefulness of our single-cell atlas to human spaceflight, we compared our core list of 375 DEGs across single cells in simulated microgravity to changes across all single cells from the Inspiration Four (I4) crew members. The I4 mission provides a compelling comparison since crew members spent up to three days in LEO,

**Fig. 5 | Validation of single-cell signatures identifies overlapping features of immune dysfunction in simulated microgravity and spaceflight. A** Volcano plot of DEGs from simulated uG vs.1G (25 hours) Bulk RNA-seq. Genes that are consistently upregulated across single-cell and bulk sequencing are labeled in red; genes that are consistently downregulated across the two datasets are labeled in blue. DEGs (including log2FC and p value) were calculated by the DESeq2 method; p value was determined by two-tailed Wald test and adjusted by the B-H method. Data were obtained from PBMCs from 3 male (ages 37, 22, 32 years old) and 3 female (age 27, 26, 40 years old) donors. **B** Spearman correlation of normalized counts between single-cell and bulk RNA-seq from simulated uG ($R = 0.82$, $p < 2.2e-16$) and 1G ($R = 0.8$, $p < 2.2e-16$) conditions. Two-tailed p value. **C** Venn diagram summarizing the overlapping DEGs between single-cell (SC; adj. $p < 0.05$, log2FC > |0.1|) and bulk RNA-seq (Bulk; $p < 0.05$) simulated uG vs. 1G. DEGs that are upregulated in both datasets are listed in the red box; DEGs that are downregulated in both datasets are listed in the blue box. The overlapping p value was calculated by the Fisher's Exact Test, right-tailed. **D** Volcano plot of DEGs from Flight (ISS 33 days, $n = 10$) vs. Ground ($n = 10$) mouse spleen bulk RNA-seq (GLDS-420). Genes that are consistently upregulated across single-cell human PBMCs and bulk mouse spleen RNA-seq are labeled in red; genes that are consistently downregulated across the two are labeled in blue. DEGs (including log2FC and adj. p) were calculated by the DESeq2 method; p value was determined by two-tailed Wald test and adjusted by the B-H method. **E** Venn diagram summarizing the overlapping DEGs between human PBMCs single-cell (SC; adj. $p < 0.05$, log2FC > |0.1|) simulated uG vs. 1G and the mouse orthologous DEGs from Flight vs. Ground spleen bulk RNA-seq (GLDS-420; $p < 0.05$). DEGs that are upregulated in both datasets are listed in the red box; DEGs that are downregulated in both datasets are listed in the blue box. The overlapping p value (5.4e-14) was calculated by the Fisher's Exact Test, right-tailed. **F** Heatmap of overlapping DEGs between human PBMCs simulated uG vs 1G and the I4 mission ($n = 4$) post-flight (R + 1) vs preflight (L-44) dataset. Both datasets are single-cell RNA-seq with DEGs defined by adj. p value < 0.05 and log2FC > |0.1|. Genes that are consistently upregulated across single-cell human PBMCs and I4 datasets are labeled in dark red (left). Genes that are consistently downregulated across datasets are labeled in dark blue (right). Genes that significantly overlap, but show reversal in their expression directions are labeled in gray. The overlapping p value (1e-200) was calculated by the Fisher's Exact Test, right-tailed. **G** Heatmap of IPA canonical pathways enriched from DEGs between human PBMCs SC (single-cell RNA-seq uG vs 1G) and I4 ($n = 4$, R + 1 vs L-44). Enriched pathways have B-H adjusted p values < 0.05 (−log10(adj. p)>1.3). Red indicates a predicted activation in pathways, whereas blue indicates a predicted inhibition in pathways.

a timeline not too different from our 25-hour time point. Moreover, I4 gene lists were also generated by single-cell sequencing, making it a comparable technology for our analysis. However, it is important to note that the I4 datasets contain a few important caveats. First, the altitude flown by I4 crew (585 km/364miles) predispose the astronauts to higher radiation exposures than what would typically be experienced on the ISS (408 km/254miles altitude). Additionally, the I4 datasets were derived from PBMC gene expression comparisons between post-flight (1 day after Return/R + 1 in our case) vs preflight (44 days before launch/L-44 in our case). Since the changes from the I4 single-cell data represent changes encompassing effects of spaceflight, plus return to the ground, including short-term exposures to hypergravity, and one day of return to 1 g gravity (all which manifest as increased gravity exposure to inflight conditions), we considered overlapping immune cell genes in either direction on return to be gravity sensitive genes. Remarkably, despite these caveats, we found a very robust overlap of nearly 60% of DEGs in PBMCs in simulated microgravity (210/375 = 56%) to be also significantly altered across all immune cells in the I4 mission (Fig. 5F, Supplementary Data 13). Of these significantly altered genes across I4 data, 122 were altered in the same direction as I4 data, and 88 in the opposite direction. To gain a better understanding of the pathways and mechanisms impacted by gravity and spaceflight in the immune system, we compared pathways between simulated microgravity and the entire I4 dataset (Fig. 5G, Supplementary Data 14, and Supplementary Fig. 16A). While we consider all of these pathways to be potentially gravity sensitive, we considered pathways altered in the opposite directions to be potentially acutely sensitive to gravity, while those pathways altered in the same direction likely take longer to normalize from a microgravity environment upon return.

Both simulated microgravity and the I4 mission pathway results indicated reduced T-cell effector subset development, reduced oxidative phosphorylation, and increased pathways associated with innate immunity (e.g., Coronavirus pathogenesis, FcR phagocytosis in monocytes, cytokine storm, chemokine signaling, ROS production in macrophages), as well as hypoxia and glycolytic metabolism (HIF1α signaling) and cell stress (e.g., sirtuin signaling). Interestingly, the return to gravity seen in the I4 mission reversed reductions in natural killer cell signaling and reversed pathways linked to poor adaptive immunity like IL-15 signaling, suggesting these pathways may be sensitive to acute changes in gravity (Fig. 5G, Supplementary Data 14). From the I4 dataset, we also noticed a consistent reduction in ribosomal subunit genes in the I4 data (Fig. 5F), which might be reflective of a stress response and reduced protein translation upon return to earth), and only some of these genes were reduced in simulated

microgravity. Consistently, there was a marked reduction in EIF2 signaling in the provided I4 DEGs. Across all pathways, regardless of direction, we noticed many pathways pertaining to the cytoskeleton or to a mechanical extracellular environment (e.g., fibrosis, RAC, Rho family GTPases, RHOA, integrin signaling, leukocyte extravasation signaling, healing signaling etc.) to be altered by simulated microgravity or by spaceflight on immune cells.

Next, to look for overlapping genes relevant to longer exposures to human spaceflight, we compared our core simulated microgravity signature of immune cells against available data from the JAXA Cell-Free Epigenome study in LEO (GLDS-530) and the Twins study[17]. For the JAXA mission, we had access to cell-free RNA data, which can sometimes give insight into changes to PBMCs, amongst other cells[29]. During this mission, blood was sampled from 6 astronauts, with data pooled into a single count, at days 5, 30, 60, 120 post launch. Given that cell-free RNA is not a fully ideal comparison to RNA-seq from isolated PBMCs, we focused only on the two early time points, 5 days and 30 days post launch, because we saw significant overlap between our single-cell data in both the three-day I4 mission and in the 33-day GLDS-420 dataset. Thus, we compared 5 days and 30 days in-flight vs preflight differentially altered cell free RNA signatures for any possible overlap with our single-cell data in simulated microgravity. While we did not observe much overlap in our core 375 immune gene signature at 5 days (less than 10 genes), we did see significant overlap by 30 days (42/375 = 11.2% overlap in the same direction) (Supplementary Fig. 16B). Interestingly, it was observed that cell-free RNA levels generally decreased across most genes in flight. Consequently, we hypothesized that identifying genes exhibiting increased expression could be particularly important for identifying over-represented processes. Remarkably, we did note that the most significantly elevated gene at 30 days in-flight vs preflight was *cdc42*, a key modulator of the cytoskeleton, as well as *dynll1* a dynein gene that was also upregulated in our single-cell analysis.

We next reclustered our single-cell data (Supplementary Data 15) to compare the DEGs in our equivalent single-cell populations with those obtained from sorted CD4⁺ T cells, CD8⁺ T cells, B cells, and lymphocyte-depleted immune cells from the NASA Twins study, which compares in-flight vs ground twin control[17]. The Twins study provides intriguing data on the impact of LEO on the immune system, but has caveats in that exposure to LEO was calculated in only a single individual through bulk RNA-seq, and at multiple time points across one full year in space, a different duration than our shorter gene sets. Nonetheless, compared to our reclustered CD4⁺, CD8⁺, and CD19⁺ gene sets, we found significant overlap in some genes comparing the effects of simulated microgravity to spaceflight (Supplementary Fig. 17A−C).

Across multiple cell types, we saw changes in genes involved in redox regulation (e.g., reduced *txnip*), and in genes involved in interferon responses (e.g., reduced *stat1* and *gbp5*). Interestingly, we saw significantly reduced cold shock gene, *cirbp*, a similar functioning gene to *rbm3*, in B cells in space. In our lymphocyte depleted (i.e., myeloid) recluster, we saw a large and highly significant overlap of about 163 genes with the Twins lymphocyte-depleted bulk RNA-seq data (Supplementary Fig. 17D). Many of the overlapping genes induced by simulated microgravity or spaceflight included genes involved in innate immunity and inflammation (e.g., *il1b*, *s100a12*, *thbs1* etc.), the cytoskeleton (*rhoq*, *rhou*), and hypoxia signaling (e.g., *hif1a*). Some interesting downregulated genes in both datasets in myeloid cells included again *gbp5*, *cirbp*, *txnip* like seen in T and B cells from the Twins study. We also noted a number of overlapping downregulated genes in antigen presentation (e.g., *tap1*, *tap2*, *hla-e*, *hla-dp1a*, etc). IPA analysis on this data largely captured the increases in innate immune inflammatory pathways, including increases in fibrosis signaling, IL-6 signaling, acute phase response, cytokine storm, and HIF1α signaling seen across some of our previous datasets (Supplementary Fig. 17E, Supplementary Data 16). Overall, these data enforce the idea of classically activated basal myeloid inflammatory changes in microgravity and spaceflight.

Given that many of our altered pathways in simulated microgravity involved predicted mitochondrial dysfunction and/or the cytoskeleton, we used Airyscan super-resolution confocal microscopy to characterize immune cell mitochondrial and actin morphological networks to look for alterations in simulated microgravity compared to 1G controls. Interestingly, while 25 hours of simulated microgravity did not alter mean cell area across PBMCs, it did alter actin granularity parameters, as well as intensity and variance, consistent with cytoskeletal changes in acute simulated microgravity (Fig. 6A–C), though these differences are mostly subtle to the naked eye. Using three-dimensional (3D) super-resolution imaging, 25 hours of simulated microgravity did not alter cell or nucleus volume, or nucleus shape, but it increased mean cell surface area and actin spike length, and decreased sphericity of the cells across PBMCs (Fig. 6A, B, D, Supplementary Fig. 18A, B). Remarkably, 1G immune cells and simulated microgravity immune cells demonstrate unique spectral changes to actin rearrangement post TLR stimulation, such that TLR stimulation resulted in a different pattern of actin granularity spectral change in 1G compared to stimulation in simulated microgravity (Fig. 6C). The effect of microgravity on the cytoskeleton in unstimulated immune cells was similar to the effect of TLR activation in 1G. During TLR stimulation in simulated microgravity, immune cells followed a unique dynamic actin rearrangement pattern, potentially even reversing the pattern observed in 1G with TLR stimulation. These results suggest that simulated microgravity itself may induce immune cytoskeleton alterations, which may mimic aspects of TLR ligation on the cytoskeleton. Short-term exposure to simulated microgravity showed some increases in the mitochondrial MitoTracker Red staining intensity and variance in the unstimulated conditions, without changes to fiber length, size, or volume (Supplementary Fig. 18C).

Since we detected morphological changes to the actin network, as well as noting changes in multiple altered cytoskeleton-related pathways across multiple datasets, including in the pathways "Signaling by Rho Family GTPases" or "regulation of actin-based motility by Rho", we next sought to screen for active GTP-bound Rho GTPases, Rac1, RhoA, and Cdc42, using G-LISA technology[30] across further batches of isolated paired PBMCs. After 25 hours of simulated microgravity, regardless of stimulation conditions, we saw elevated levels of active GTP-bound Cdc42, consistent with cytoskeleton mobilization and the increase in actin spikes (indicative of filopodia)[31] observed due to simulated microgravity (Fig. 6E). Active GTP-Rac1 was not altered at baseline in simulated microgravity, though showed a trend to induction with TLR7/8 stimulation (Supplementary Fig. 18D, E). Levels of

active GTP-RhoA were low in our samples, but trended lower in simulated microgravity without stimulation, and higher under stimulation, analogous to our single-cell data predictions (Supplementary Fig. 18D, E). Overall, these data suggest that simulated microgravity changes some Rho GTPase activity consistent with our transcriptional data, though ultimate impacts on cytoskeleton shape, variance, and dynamics likely involve additional contributing factors, including possibly other Rho GTPase family members not assessed.

We next sought to investigate our core signature of reduced IFN signaling elicited in microgravity across immune cells (Fig. 6F, Supplemental Fig. 19A, B). Specifically, we assessed whether reduced interferon signaling was due to reduced local production of interferons. Supernatants from 25 hours unstimulated or 9 hours R848 simulated (25 hours total culture) PBMCs were assessed by ELISA for total IFNα (detecting 12 IFNα subtypes), and IFNγ. Simulated microgravity significantly reduced both IFNα and IFNγ secretion with TLR7/8 stimulation. At baseline, the levels of these cytokines were low, and variable, and thus not significantly different between 1G and simulated microgravity. These findings point to reduced production of IFNs in simulated microgravity, at least under TLR stimulation, as measured by ELISA, potentially as one contributing mechanism to reduced interferon signaling observed at the transcriptional level.

Finally, to functionally validate how simulated microgravity impacts overall immune cell cytokine production, with and without TLR7/8 stimulation, across many cytokines simultaneously, we performed a 48-plex Luminex assay on cytokines secreted by PBMCs from 12 donors (Supplementary Fig. 19C, D). Consistent with our single-cell and bulk RNA sequencing data, simulated microgravity was associated with increased or trending increases in mainly innate/monocyte immune cell-derived inflammatory cytokines and chemokines (e.g., IL-6, IL-8, IL-12p40, CCL4), coupled to a reduction in cytokines that associate with T cell activation or proliferation (e.g., IL-2, IL-7, IL-15). Concurrently, the Luminex results showed a significant IFNγ and a trending IFNα2 reduction upon TLR7/8 agonist stimulation in simulated microgravity, consistent with our above ELISA data (Fig. 6F). IL-1, commonly induced in our sequencing data, also appeared elevated in simulated microgravity, though it exhibited high variability, precluding significance in the unstimulated state. In the stimulated state, IL-1β was significantly increased in simulated microgravity by Luminex analysis. Given the overlapping similarities between cytokines in the Luminex data and sequencing data for IL-1β, IL-6, and IL-8, we further assessed these cytokines by ELISA validation. Both IL-6 and IL-8 showed significant or near-significant increases by ELISA in simulated microgravity, while IL-1β demonstrated a trending increase (Fig. 6G, Supplementary Fig. 19E). Upon stimulation, simulated microgravity further facilitated near-significant increases in IL-1β and IL-8 as validated by ELISA (Supplementary Fig. 19F).

To better understand how certain cell populations respond to TLR7/8 stimulation in simulated microgravity, we further validated key cytokines, IL-1β, IL-6, and IFNγ, by intracellular flow cytometry in monocyte, NK, and T cell subsets exposed to simulated microgravity compared to 1G conditions (Supplementary Figs. 20, 21). Consistent with Luminex and ELISA data, we saw increased IL-1β production across all characterized monocyte populations (Supplementary Fig. 21A, B). Interestingly, despite no overall differences in IL-6 by Luminex or ELISA in simulated microgravity to TLR7/8 stimulation, we still detected significant increases in a subset of monocytes only, as well as in NK cells (Supplementary Fig. 21C, E). Despite increased cytokine production, we did not detect increases in the activation marker, HLA-DR, in monocyte populations (Supplementary Fig. 21D). NK cells also showed a reduction in the proportion producing IFNγ, as well as reduced proportions of expression in the activation marker, CD69, and degranulation marker, LAMP-1, consistent with reduced functionality and response to stimulation in simulated microgravity (Supplementary Fig. 21F, G). T cell subsets were less altered, though we

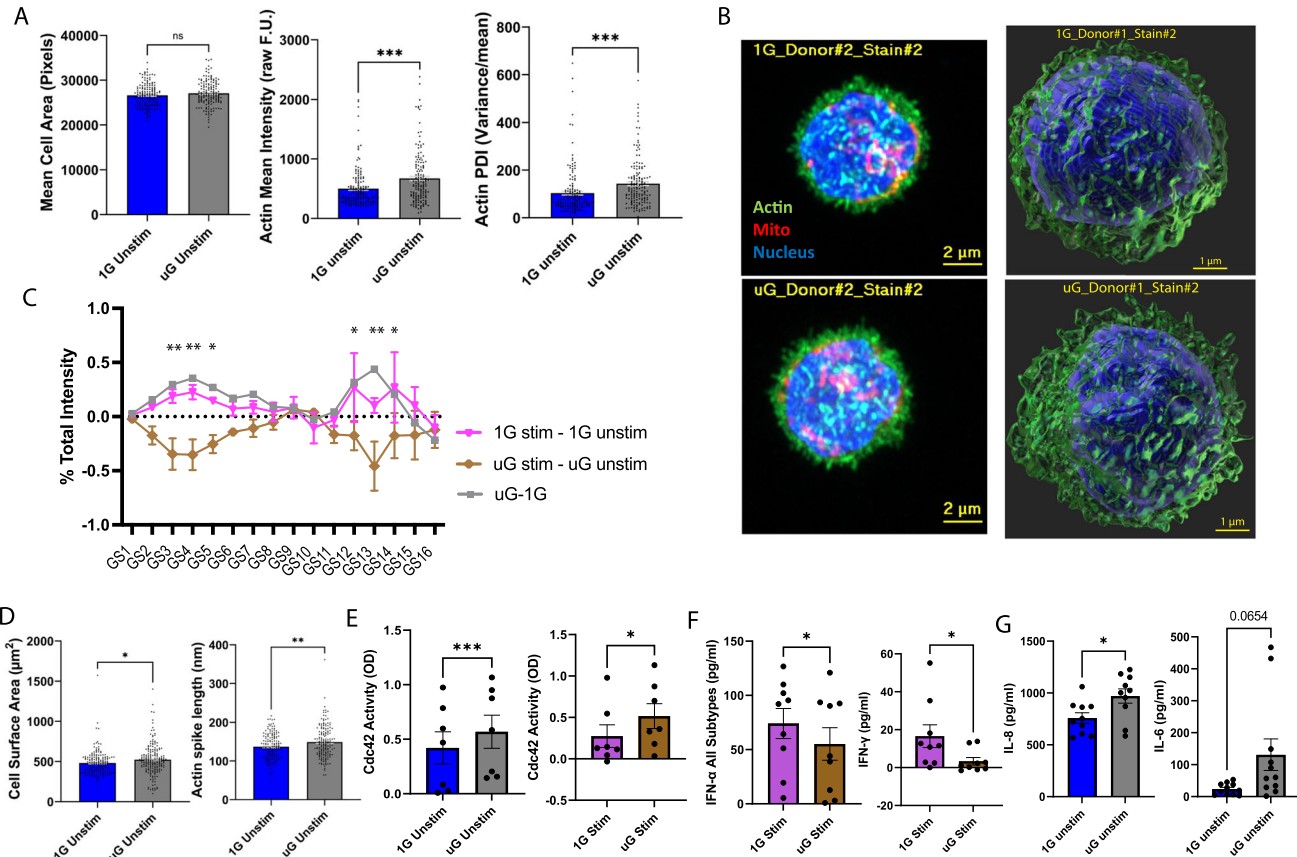

**Fig. 6 | Simulated microgravity induces distinct modifications to immune cell cytoskeletal morphology and cytokine production. A** Super-resolution microscopy analysis of actin in 2D for cell area (left), intensity (middle), and texture as punctate over diffuse index (PDI, variance/mean, right) between 25 hours of 1G or simulated uG. Dots represent individual PBMCs ($n = 159$ cells for 1G, and $n = 154$ cells for uG) from 4 independent donors. Donors were male (25 years old), and females (35, 38, and 46 years old). One outlier for actin intensity and actin PDI from each condition is removed based on Grubbs' test. Two-tailed Welch's $t$ test was used for all comparisons. ***$p \le 0.001$. Data are plotted as mean $\pm$ standard error of the mean (SEM) and source data are provided with this paper. **B** Representative super-resolution microscopy images (2D left, 3D right) of PBMCs from 1G and simulated uG (25 hours) from donor 1 (35 yr F) and donor 2 (25 yr M; 2 of total 4 donors from **A** are shown here). 3D images better highlight changes to overall cell shape and actin protrusions in simulated uG. Scale bar = 2 µm and 1 µm, respectively. **C** Sixteen-channel granularity spectrum measurement of PBMCs stimulated with TLR7/8 agonist (9 hours stimulation, 16 hours conditioning prior to stimulation) from 1G (pink line) and uG (brown line) minus the corresponding unstimulated cells (25 hours total culture). The effect of simulated microgravity on unstimulated granularity spectrum is plotted in gray. Asterisks compare pink vs brown lines only. $P$ values generated from unpaired two-tailed $t$ test. $n = 3$ donors tested from **A**, 35-year-old female sample was not used. **$p \le 0.01$, *$p \le 0.05$. Data are plotted as mean $\pm$ SEM, and source data are provided with this paper. **D** Super-resolution

microscopy analysis of 3D actin surface area (left, 1G $n = 179$ cells and uG $n = 194$ cells) and actin spike length (right, 1G $n = 162$ cells and uG $n = 165$ cells) between 25 hours of 1G or simulated uG. Dots represent individual PBMCs from 4 independent donors. Donors were male (25 years old), and females (35, 38, and 46 years old). Two-tailed Welch's $t$ test was used to calculate $p$ values. *$p \le 0.05$, **$p \le 0.01$. Data are plotted as mean $\pm$ SEM and source data are provided with this paper. **E** G-LISA levels of active GTP-bound Cdc42 in PBMCs either unstimulated (25 hours) or treated with TLR7/8 agonist (9 hours + 16 hours conditioning) from 1G and simulated uG. $n = 7$, donors were male (25 years old), and females (38, 46, 25, 27, 26, and 40 years old). Two-tailed paired parametric $t$ test was used to calculate $p$ values, *$p \le 0.05$, ***$p \le 0.001$. Data are plotted as mean $\pm$ SEM and source data are provided with this paper. **F** ELISA levels of secreted IFNs by PBMCs treated with TLR7/8 agonist (16 hours conditioning + 9 hours stimulation) from 1G and simulated uG. $n = 9$, donors were male (36 years old), and females (33, 25, 38, 46, 27, 25, 26, and 40 years old). Two-tailed paired parametric $t$ test was used, *$p \le 0.05$. Data are plotted as mean $\pm$ SEM, and source data are provided with this paper. **G** ELISA level of secreted ILs by PBMCs exposed to 25 hours simulated uG and 1G. $n = 10$ for IL-8 and $n = 11$ for IL-6, donors were females (32, 25, 38, 46, 25, 27, 26, 40 years old) and males (36, 33, 26 years old); 38-year-old female sample was not used for IL-8. Two-tailed paired parametric $t$ test was used, *$p \le 0.05$. Data are plotted as mean $\pm$ SEM and source data are provided with this paper.

still detected near significant or significant reductions in the proportions of CD4$^+$ and CD8$^+$ central memory T cells expressing activation marker, CD69, and in effector memory CD4$^+$ T cells expressing proliferation marker, Ki67 (Supplementary Fig. 21H). Taken together with our ELISA and Luminex data, these findings demonstrate that simulated microgravity, alone or in the presence of TLR7/8 agonist, can functionally alter cytokine production across immune cells. In general, consistent with sequencing data, the features demonstrate monocyte inflammatory function coupled to impaired T cell and NK cell functionality in simulated microgravity. Thus, changes in cytokine signaling observed in simulated microgravity may occur at least in part to changes in upstream cytokine production.

**Reversing simulated microgravity effects on the immune system**
We have characterized multiple genes and pathways altered by simulated microgravity in the immune system; however, whether there are specific drugs or supplements that can directly target microgravity effects on immune cells is poorly characterized. Thus, we have utilized an in-house compound-gene interactome machine learning technology (Gene Compound Enrichment Analysis, GCEA), building on the HyperFoods model[32], for the identification of drugs and food supplements that significantly map to altered genes in a dataset. Overall, our pipeline assesses >2 million interactions between genes, drugs, and foods, across DrugBank, LINCS, and FoodDB[32] (Fig. 7A). Using these algorithms across our core signature

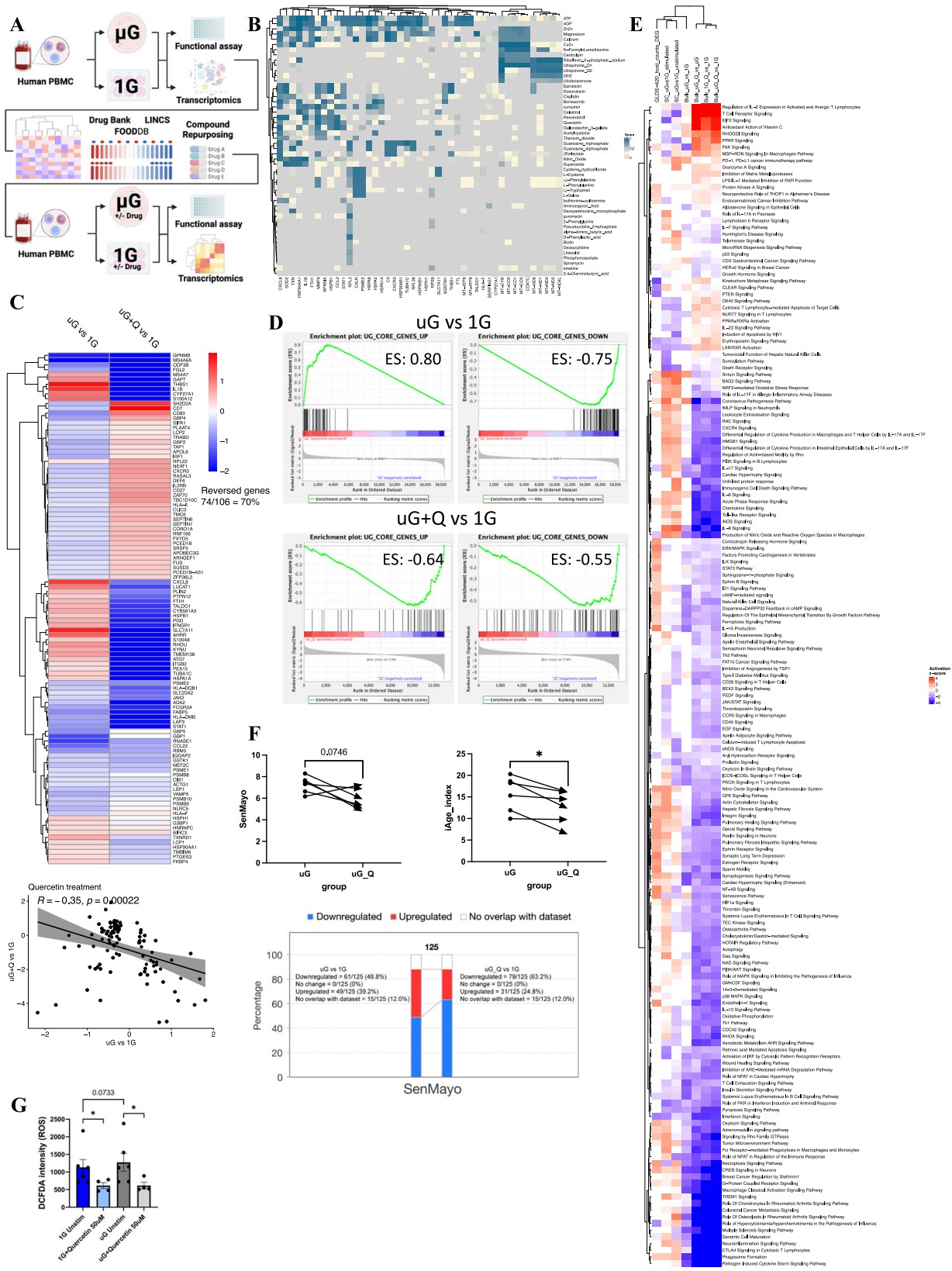

of 375 DEGs altered by simulated microgravity across the immune system, we identified 115 compounds with adj $p < 0.05$, and 474 compounds with $p < 0.05$ that significantly map to our signature (Fig. 7B and Supplementary Data 17). Figure 7B shows the top 50 most significantly overlapping compounds to enriched DEGs. We next chose one compound, quercetin, based on its widespread

availability for future travelers to space, and for its prominence as an anti-aging supplement, to validate whether it can reverse transcriptional insults of microgravity on the immune system. PBMCs (donors from the Fig. 5A cohort) were subjected to 25 hours in simulated microgravity, with or without quercetin (50 μM), for bulk RNA-seq analysis.

**Fig. 7 | Reversing simulated microgravity effects on the immune system with quercetin. A** Pipeline of microgravity and gene interacting compounds from discovery to validation. **B** Heatmap of top 50 simulated uG altered gene to compound interaction candidates. Compounds are listed on the right, and the predicted interacting genes are listed at the bottom. The color indicates the STITCH confidence score for compound-gene interaction. **C** Quercetin (50 μM) reverses the core gene expression signatures in simulated uG (25 hours). Log2FC levels of 106 core DEGs from simulated uG vs. 1G are plotted side-by-side to quercetin-treated uG vs. 1G in the heatmap. Red indicates positive log2FC, and blue indicates negative log2FC. 70% of the genes are reversed after quercetin treatment. The scatter plot below shows a negative association (Pearson correlation $R = -0.35$, $p < 0.001$) between the log2FC levels of the 106 core genes from simulated uG vs. 1G and quercetin-treated uG vs. 1G. **D** Gene set enrichment analysis (GSEA) shows the reversal effect of quercetin on the 106 core DEGs plotted in **C** heatmap. Quercetin treatment inverts the enrichment score (ES) in the upregulated core genes (from 0.8 to −0.64) and increases the ES of the downregulated core genes (from −0.75 to −0.55). All $p$ values are <0.0001. **E** IPA Canonical pathways altered by quercetin. Heatmap plots the comparison of quercetin-treated samples against non-treated

bulk RNA-seq controls, single-cell TLR7/8 agonist stimulated and unstimulated samples, and GLDS-420 mouse spleens in space. Red indicates a predicted activation in pathways, whereas blue indicates a predicted inhibition in pathways. The datasets were clustered by quercetin treatment into 2 major groups via complete linkage hierarchical clustering method. **F** Quercetin reduces senescence and age-associated inflammatory gene outputs. Both SenMayo scores and iAge index are reduced in the quercetin-treated group with $p$ value of 0.0746 and 0.0268 respectively. Compared with the untreated group, quercetin downregulates more senescence-related and age-associated inflammatory genes (from ↓48.8% to ↓63.2%). $n = 6$, donors were 3 males (age 37, 22, 32 years old) and 3 females (age 27, 26, 40 years old). Two-tailed paired $t$ test, *$p \le 0.05$. **G** Quercetin (25 hours treatment) reduces ROS levels measured by 2′,7′-dichlorofluorescin diacetate (DCFDA) assay. $n = 6$ for 1G vs uG, donors were males (32, 37, and 38 years old) and females (34, 32, 37 years old). 34 yrs and 32 yrs female samples were not treated with quercetin, resulting in $n = 4$ for comparisons between 1G vs 1G+Quercetin and uG vs uG+ Quercetin. Two-tailed paired parametric $t$ test was used, *$p \le 0.05$. Data are plotted as mean ± SEM and source data are provided with the paper.

Remarkably, at the gene level, quercetin reversed the direction of expression of 70% (74/106) of the 106 genes (Fig. 5C) core signature generated as significantly overlapping genes between single-cell and bulk RNA-seq validation (Fig. 7C). Reversal of gene expression was significant by correlation analysis (Fig. 7C) and demonstrated by GSEA enrichment plot (Fig. 7D). IPA pathway analysis was then performed to characterize pathways altered by quercetin, and compared against non-treated bulk RNA-seq controls, as well as pathways altered across our other 3 major datasets, including single-cell sequencing and GLDS-420 spleens in space. Overall, pathway analysis across all datasets showed consistent impacts of simulated and actual microgravity on pathways essential for optimal immunity (Fig. 7E, Supplementary Data 18). Some of the most consistently induced pathways in simulated microgravity and/or space included "coronavirus pathogenesis pathway" (linked to innate immune activation), acute phase responses, leukocyte extravasation signaling, IL-6 signaling, BAG2 signaling (linked to heat-shock proteins and proteostasis), sirtuin signaling, and to a lesser extent "regulation of actin-based motility by Rho", RAC signaling, PKA signaling, and oxidative stress response. Major pathways attenuated by microgravity were linked to immunity, including antimicrobial immunity, pyroptosis signaling, as well as "interferon signaling" (including PKR in IFN induction). Other reduced pathways across most datasets included reduced nuclear receptor activation (including LXR/RXR, PPAR, AHR) and reduced T cell NUR77 (activation) signaling. Interestingly, we noted some genes were consistently reduced across all datasets, though were not properly represented in pathway analysis. The most striking of these genes is *rbm3*, a cold-shock protein, which was significantly reduced in all four of the microgravity datasets, as well as in the I4 and JAXA mission (30-day timepoint). *Rbm3* was also reduced in the Twins study inflight data across all sorted immune cells, though not reaching significance.

Administration of quercetin in simulated microgravity could reverse many of the altered transcriptional signatures elicited by simulated microgravity on the immune system (Fig. 7E). Some of the major pathways it could reverse include "regulation of actin-based motility by Rho", leukocyte extravasation signaling, RAC signaling, LXR/RXR, PPAR signaling, NUR77 signaling in T cells, "coronavirus pathogenesis" (innate immunity), acute phase response, fibrosis, IL-6 signaling, amongst others. Though quercetin has gained prominence for its senolytic properties[33], our results show that reducing senescence pathways was only one of many (approximately 174) pathway effects this compound has on immune cells in simulated microgravity (Supplementary Data 18). Nonetheless, in simulated microgravity, quercetin could reduce senescence and age-associated inflammatory

gene outputs, as demonstrated by reductions in both the SenMayo and iAge index scores (Fig. 7F). These changes occurred for the most part by downregulating inflammatory genes.

Despite the marked transcriptional reversal in simulated microgravity observed with one compound, quercetin failed to reverse reductions in interferon signaling, a major hallmark of microgravity on immune system dysfunction from our data. Other studies have also linked microgravity and spaceflight to mitochondrial dysfunction and ROS production[10,34,35]. In this regard, quercetin also showed a robust capacity to reduce ROS levels after 25 hours of simulated microgravity (Fig. 7G), though ROS was only marginally increased as a trend by simulated microgravity itself after 25 hours, likely due to the expression of endogenous antioxidant systems at this timepoint[34]. Consistently, we also observed increased oxidative stress responses, such as NRF2-mediated or sirtuin signaling in many of our transcriptomic datasets by IPA analysis (Fig. 7E).

## Discussion

Immune dysfunction during spaceflight is an important health risk, and manifests primarily as increased vulnerability to opportunistic infections, including latent viral reactivation[3]. Latent viruses can reactivate on both short- and long-term spaceflights, and commonly involve herpes viruses (HSV1, EBV, CMV, VZV)[12–15]. Astronauts also experience heightened skin sensitivity reactions[3,13,36], and this mechanism was thought to be related to a possible Type 2 immune bias in space[7,16,36]. Recent work in simulated microgravity has also shown reduced JAK/STAT signaling in CD8⁺ T cells, coupled to increased pSTAT5 signaling in Tregs. Despite these important advances, major mechanisms explaining these phenotypes of immune dysfunction, in simulated microgravity have remained unclear.

We have identified numerous core pathways and genes altered across human immune cells in simulated microgravity, with validation against datasets of humans in LEO, as well as spleens from mice flow on the ISS. Overall, we noted changes consistent with basal innate immune cell inflammatory changes in simulated microgravity, coupled with distinct pathways of dysfunction in multiple immune cells. Specifically, the most consistently reproduced pathways impacted by simulated microgravity across immune cells in both single-cell and validation cohorts included changes to pathways and signaling linked to acute phase response signaling, Coronavirus pathogenesis, IL-6 signaling, the cytoskeleton, interferon response, pyroptosis, heat-shock, nuclear receptors, and sirtuin biology.

The link between the cytoskeleton and other pathways here may be especially relevant in immune dysfunction. Cytoskeleton dynamics are controlled by a number of factors, but small GTPases, including Ras homology (Rho) GTPases, are major orchestrators with critical impact

on immune cell function, migration, gene expression, trafficking, phagocytosis, proliferation, and antigen recognition[31]. Of note, Rho GTPases have been implicated in response to simulated microgravity in other cell types, but this connection is understudied in immune cells[37]. Across most datasets, we saw changes to Rho GTPase signaling, including individually in RAC, RHOA, or CDC42 signaling, or combined in a global "regulation of actin-based motility by Rho" pathway in IPA. While we did notice some variability between our initial unstimulated vs stimulated single-cell data in these pathways, these pathways tended to show reduced RHOA signaling without stimulation, coupled with increased RAC signaling, analogous to what we observed from the I4 crew members upon landing. The JAXA6 dataset also demonstrated *cdc42* to be the most significantly induced cell-free transcript in astronauts after 30 days in space. Pathways strongly linked to cytoskeletal remodeling, such as leukocyte extravasation, were also typically induced in most of our datasets. We also observed changes in some active Rho GTPases by G-LISA, as well as in F-actin granularity, variance, 3D surface area, sphericity, actin protrusion length, and dynamic change to TLR stimulation by super-resolution microscopy, providing further evidence for changes in actin, including possibly immune cytoskeleton alteration or dysfunction, in simulated microgravity.

Importantly, changes to the actin cytoskeleton are now being linked to the ability of an immune cell to mount an interferon response. Indeed, danger-sensing molecules like TLRs utilize Rho GTPases to facilitate IFN responses[38,39], or antiviral sensors can directly modulate actin rearrangement[40]. One example is the PKR antiviral response, which was consistently downregulated by simulated microgravity in our datasets. In this system, PKR binds gelsolin to enforce basal innate immune defense, though upon viral sensing, PKR dissociates from gelsolin, leading to the severing of actin, and activation of RIG-I-like receptor (RLR)s signaling and interferon response[41]. Other antiviral sensors like RIG-I directly bind F-actin in resting cells, and then relocalize to the mitochondria via actin rearrangements on viral infection, to induce type 1 IFN[42,43]. In single-cell data, reduced interferon signaling without stimulation was seen mainly in monocytes, linking it to innate immunity, though with TLR7/8 stimulation, reduced interferon signaling was seen across many cells, including most T cell subsets, and NK cells, displaying the broad importance of this pathway across most immune cells to microgravity. In simulated microgravity, we saw reduced IFNα production by ELISA with stimulation, so we cannot rule out the possibility that the reduced IFN signaling seen in simulated microgravity starts with reduced capacity for IFN production in some conditions, in addition to potential defects in downstream signaling itself. While we have focused on type 1 IFN signaling, some reduced interferon responses are also linked to reduced signaling from the IFN gamma receptor (IFNGR). Consistently, we also noted reduced IFNγ production in simulated microgravity upon TLR7/8 stimulation. Whether the cytoskeleton is needed for IFNGR clustering and signaling remains to be seen[44].

Consistent with reduced interferon signaling in simulated microgravity, we noticed a reduction in some IFN-inducible GTPase superfamily genes, namely guanylate binding proteins (GBPs) across our datasets. Various GBPs (e.g., *gbp5*) were reduced as well in the Twins study. Interestingly, GBPs, which are heavily induced by IFNγ signaling, have been shown to be critical in maintaining responses to *mycobacterium tuberculosis*[45,46], and we see reactivation of similar bacteria (in addition to some retroviruses) in simulated microgravity after as little as 25 hours of exposure. GBPs and associated IFN responses also help direct inflammasome activation and pyroptosis (an inflammatory form of cell death) linked to antimicrobial defense that was consistently down in monocytes and B cells in simulated microgravity, and in nearly all immune cells in response to TLR7/8 stimulation in simulated microgravity[46,47]. Interestingly, pyroptosis and inflammasome

activation can also be directly controlled by Rho GTPases and the cytoskeleton[48,49].

Another pathway found consistently down across datasets included LXR signaling. Interestingly, LXR signaling also can promote antimicrobial defense mechanisms. Macrophage LXR has been shown to reduce bacterial infection by reducing intracellular $NAD^+$ in a CD38 manner, with mechanistic impacts on the cytoskeleton[50]. Whether $NAD^+$ levels fall in microgravity remains to be seen, though we did see an interesting increase in sirtuin signaling across datasets, including in the I4 mission. Sirtuins may be functioning to counter acute oxidative stress in microgravity[34,51]. We also saw reduced oxidative phosphorylation transcriptional signatures across all unstimulated immune cells in simulated microgravity. Altered metabolite levels (and possibly ROS) from impaired oxidative phosphorylation might also contribute to HIF1α stabilization[52] as observed in some of our simulated microgravity and spaceflight datasets. Reduced oxidative phosphorylation can be associated with increased glycolysis in immune cells[53], fueling "M1-like" pro-inflammatory changes in macrophages, potentiating NF-κb signaling, acute responses and IL-6 or IL-1 release, cytokines frequently induced in microgravity. Consistently, we did notice a preferential enrichment of predicted "macrophage classical activation" signatures across our gene sets in the Twins study.

Interestingly, we noted frequent increases in heat-shock genes, coupled to increased associated BAG signaling pathways across antigen-presenting cells (monocytes, B cells, and DCs), as well as in double negative T cells. Heat-shock expression may be reflective of altered proteostasis in simulated microgravity[10], and may be required for adaptation to mechanical unloading in some cells[34], though this may also be linked to higher temperatures. Across all gene sets, we noticed a reduction in the cold shock gene, *rbm3*, which was reduced in nearly all immune cells in our single-cell data. Increased heat-shock coupled with reduced cold shock genes raises the possibility of higher intracellular temperatures directly induced by microgravity, but whether microgravity, or associated increase in cytokines or binding partners such as IL-1ra, directly induce the observed "space fever" in astronauts requires further insight[54]. Interestingly, we did notice a number of significant IL-1 ligands in innate cell to T cell interactions in our microgravity Interactome, highlighting the possible importance of this cytokine family and downstream interacting molecules.

Pertaining the aforementioned skin lesions in astronauts, it has been postulated that persistent skin hypersensitivity reactions in some crew members may be linked to allergic responses[7,16]. While analysis of our datasets cannot rule out this possibility, we did not observe increased Th2 signatures across our simulated microgravity systems, or with our specific gene sets validated across the I4 mission. We also saw inconsistent changes to IL-23 and IL-17 signaling across our datasets, though these cytokines are known contributors to skin disease[55]. While the root causes of such lesions cannot be inferred from our data analysis, it likely involves changes in the crosstalk with the skin microbiome, in addition to intrinsic immune cell abnormalities. Interestingly, we did see reduced aryl hydrocarbon receptor (AHR) signaling in most of our datasets, especially in CD14 monocytes and conventional type 2 dendritic cells, raising the idea of reduced AHR signaling in space to contribute to skin lesions[56,57]. However, we saw AHR signaling was enriched in the Twins study gene set result and so more experimentation is needed to tease out a possible role for the AHR in astronaut skin lesions.

Using in house machine learning algorithms, we identified numerous compounds mapping to microgravity's transcriptional response to the immune system. This algorithm focuses on the strength of interaction and does not specify direction. However, we tested one of the most significantly interacting compounds, the flavonol, quercetin, for its ability to reverse transcriptional changes to simulated microgravity on the immune system, and found that it could reverse approximately 70% of altered core genes. Of note, quercetin

reversed numerous pathways, including core pathways such as reduced nuclear receptor activation, sirtuin signaling, Coronavirus pathogenesis pathway, associated acute phase responses and IL-6 signaling. Quercetin also showed impact on the cytoskeleton, favoring a freezing of pathways linked to its mobility in microgravity, by reducing genes associated with Rho GTPase signaling (e.g., reducing RAC, RHOA and CDC42 signaling), and boosting RHOGDI signaling. Despite these changes, quercetin was unable to revert the core immunosuppression pathway of reduced interferon responses. However, since actin skeleton mobility is needed to induce an IFN response in many instances[40], too much interference could contribute to a persistent lack of IFN signaling here, and might represent a further mechanism of immune suppression mediated by quercetin that requires more study. Interestingly, after 25 hours in simulated microgravity, we saw variable results on the induction of senescence pathways, though quercetin markedly reduced senescence associated transcripts in our data. Thus, while quercetin acts in part through its senotherapeutic mechanisms[32], the large breadth of additional other pathways suggests multiple beneficial modes of activity for immune modulation in microgravity.

Our data support a hypothetical model where microgravity alters forces sensed by immune cells, leading to changes in the actin cytoskeleton, and nuclear receptor signaling, coupled to changes in core pathways in space such as mitochondrial dysfunction and oxidative stress. Recent work in other cells, such as endothelial cells, has identified cytoskeletal abnormalities as a key feature of simulated microgravity that drives autophagy and a reduction in mitochondrial mass after 72 hours of exposure[35]. Our datasets would support some of these findings. Combined, these pathways would contribute to reduced oxidative phosphorylation and associated basal inflammatory processes, as well as reduced viral sensing pathways, associated reduced interferon responses and altered pyroptosis capability. Reduced interferon responses and signaling, impact both innate cells like monocytes and NK cells, as well as adaptive cells like T cells. Such changes could cumulate in viral or mycobacterial reactivation in microgravity. These processes would also be complemented by the psychological and physiological stresses of spaceflight, which also may independently associate with viral reactivation[12,14,58].

Thus, an important future direction of research is to address whether altered cytoskeleton, or associated reductions in interferon related gene products, including GBPs, are actively driving the reactivation of mycobacteria and latent viruses that we see in simulated microgravity. Another related important avenue of future research is to better understand how changes to force action on immune cells link to the mitochondria dysfunction hallmark of spaceflight and metabolic alterations, and to map such changes to immune cell metabolism as has been done to other mechanical forces[19,59]. Moreover, the immune system in spaceflight is also under the influence of changing pressure gradients, such as increased pressures in parts of the upper body[60], coupled to lower central venous pressure and potentially altered shear force[61–63]. These changes may also contribute to altered immune function, including the increased IL-1 and IL-6, two cytokines sensitive to external force cues, seen in spaceflight, with potential for intersecting signaling nodes across mechanotransduction and other pathways such as mitochondrial dysfunction within immune cells[19]. More work is needed to understand how these additional changes in force impact innate and adaptive immunity during spaceflight and how they interact with the effects of microgravity.

Finally, despite the multiple proposed mechanisms resulting from this study, much of it was produced using simulated microgravity as a model system, which has its own caveats[18], though we did identify core reproduced pathways from spleens in mice on the ISS and some overlap with the I4 mission and Twins study. Further studies applying single cell-omic technologies to immune cells during in-flight missions will no doubt provide answers to refine the proposed mechanisms of immune dysfunction in space. As well, integrating more in-depth

chromatin and analytical approaches, such as ATAC-seq, can help detail the regulatory changes that might be underlying these immune phenotypes[64]. Overall, our current work provides a resource to better understand "astroimmunology", in particular how and why the immune system changes in simulated microgravity and spaceflight. These results also provide opportunities to develop countermeasures that will help normalize immune cell function in microgravity and spaceflight.

## Methods

Studies were conducted under the supervision and in accordance with ethical guidelines of the Buck Institute for Research on Aging. In particular, protocols for the purchasing of human blood products from Stanford University are described in the next section below.

### Human blood sample and cell culture

De-identified peripheral blood buffy coat samples were obtained from the Stanford University Blood Center under official signed contract agreements with the Buck Institute for Research on Aging, following the Stanford Blood Center's Certification of Human Subjects Approval for minimal risk research-related activities (IRB eProtocol# 13942).

A total of 27 healthy human buffy coats between the ages of 20 and 46 were purchased from the Stanford University Blood Center. PBMCs were isolated using a Ficoll gradient method. PBMCs were counted and resuspended in complete media at $1 \times 10^6$ cells/ml (RPMI 1640, 10% Fetal Bovine Serum, 2 mM L-glutamine, 1% penicillin/streptomycin, 0.1 mM non-essential Amino acid, 1 mM sodium pyruvate, 50 µM 2-mercaptoethanol, 10 mM HEPES). To generate simulated microgravity, the cell suspension was loaded into 10 ml disposable high aspect ratio vessels (Synthecon, Houston, TX) and rotated at 15 rpm for 25 hours. For the 1G control, the cell suspension was plated in standard 6-well culture plates, as standard static culture plates or culture flasks have been shown to be comparable to static high aspect ratio vessels by others in major immunological assays[2]. 1G and simulated microgravity cultures were simultaneously placed in the same 37 C, 5% $CO_2$ incubator. To stimulate PBMCs, samples were mixed with 1 µM R848 (TLR7/8 agonist, Invivogen, San Diego, CA) after 16 hours, for 9 hours of stimulation. At the end of each experiment, the cell suspension was quickly collected, spun down at $500 \times g$, washed with phosphate-buffered saline, and used for downstream analysis. For super-resolution imaging cells were fixed before centrifuging.

### Single-cell RNA sequencing

$1 \times 10^4$ PBMCs from each condition were counted and loaded on the 10X Genomics Chromium Controller and the libraries were prepared using Chromium Next GEM Single-Cell 5' Reagent Kit v2 according to the manufacturer's protocol (10X Genomics, Pleasanton, CA). The quality of libraries was assessed using Agilent TapeStation 4200 (Santa Clara, CA), and test-sequenced on Illumina NextSeq 550 (San Diego, CA). The full sequencing was performed on an Illumina NovaSeq 6000 by SeqMatic (Fremont, CA).

### Processing of single-cell RNA-seq data

Data processing was performed using 10x Genomics Cell Ranger v6.1.2 and MTD[26] pipelines. The "cellranger count" was used to perform transcriptome alignment, filtering, and UMI counting from the FASTQ (raw data) files. Alignment was done against the human genome GRCh38-2020-A. Cell numbers after processing were: 1G unstimulated 13,304 cells, uG unstimulated 21,709 cells, 1G stimulated 16,397 cells, and uG stimulated 14,913 cells. The MTD pipeline was used to generate the single-cell microbiome count matrix from the FASTQ files.

Downstream analyses were performed in R (version 4.2.0), primarily using the Seurat R package (version 4.1.1)[65,66] and custom analysis scripts. First, we executed a quality control step that removed the cells containing >10% mitochondrial RNA and ≤250 genes/features.

The doublet cells were identified and removed from the downstream analysis by using the DoubletFinder R package (version 2.0.3)[67] with parameters PCs = 1:30, pN = 0.25, and nExp = 7.5%. To avoid the influence of hemoglobin transcripts on the analysis, we filtered out the putative red blood cells (defined by the method below) before the following process. A total of 55,648 cells remained for subsequent analysis. Raw RNA counts were first normalized and stabilized with the SCTransform v2 function (SCT), then followed by the CCA integration workflow for joint analysis of single-cell datasets. In doing so, the top 3000 highly variable genes/features among the datasets were used to run SCT; and then 3000 highly variable genes/features and the 30 top principal components (PCs) with k.anchor = 5 were used to find "anchors" for integration. The clustering step was executed by using the 30 top PCs summarizing the RNA expression of each cell with a resolution parameter of 0.8.

To identify putative cell types, Azimuth (version 0.3.2)[65] pipeline was used with the reference dataset of Human–PBMC celltype.l2. Cell type annotation results from Azimuth were validated by checking the markers of each cell type (Supplementary Data 3). Gene differential expression analyses were done by Seurat PrepSCTFindMarkers then FindAllMarkers/FindMarkers functions with MAST[68] algorithm. The pseudo-bulk analysis was conducted to find overall DEGs of uG against 1G in either unstimulated or stimulated PBMCs, using the FindMarker function with parameter min.pct = 0.005 and logFC = 0.1. To compare the stimulated and unstimulated PBMCs under uG and 1G conditions, we subtracted log2FC values of their DEGs (uG−1G). The top 50 most upregulated DEGs between stimulated and unstimulated PBMCs under 1G were used for comparison. FindConservedMarkers function was used to find DEGs that are conserved between the groups with the same parameter settings as FinderMarkers. The top 50 conserved DEGs specifically sensitive to uG were selected based on the rank of the absolute sum of log2FC values, derived separately from the sum of positive log2FC values and the sum of negative log2FC values. Rank-Rank Hypergeometric Overlap (RRHO) analysis[69,70] was performed by using RRHO2 R package (version 1.0) to compare the differential expression patterns between 1G and uG of stimulated vs unstimulated PBMCs. The ranks of the genes in the two gene lists were determined by calculating −log10(adj.$p$value)*log2FC.

### Pathway analysis

Following differential expression, Ingenuity Pathway Analysis (IPA, Qiagen) was used to discover changes in enriched pathways in each comparison. DEGs with $p$ values < 0.05 and |Log2FC|> 0.1 were incorporated into the IPA canonical pathway analysis.

### Trajectory analysis

To study the inferred trajectory of PBMC differentiation, cell trajectory analysis was performed by using the Monocle 3 R package (version 1.2.9)[71,72]. We first subsetted Seurat data to uG and 1G groups then run the functions as.cell_data_set(), cluster_cells(), and learn_graph(). Then, we ran order_cells() with the selection of cell types representing early development stages (CD4 naive, B naive, plasmablast, and HSPC) as the roots of the trajectory.

### Calculating cell scores of inflammatory aging and cellular senescence

The inflammatory aging (iAge)[24] index was calculated by the sum of the cell scores that count by multiplying normalized and transformed gene expression with the corresponding coefficient of the gene in the iAge gene set. Cellular senescence was scored using Seurat AddModuleScore function[65,73] on the SenMayo gene set[25].

### Viral and microbial abundance analysis

The output reads counts from MTD pipeline were then combined with the host reads and analyzed in R with Seurat package and other customized scripts. The relative abundance (frequency) of a virus or microbe was determined by dividing its reads count by the total reads count (host and non-host) in that sample. The classification results were further validated using a different method Magic-BLAST[74].

### APCs to T cell intercellular communication

To study the difference in intercellular communication from APCs to T cells between uG and 1G, we used nichenetr R packages (version 1.1.0)[27] to analyze cells in the dataset belonging to APCs (B cells, DCs, or monocytes) and T cell types. The "Differential NicheNet" workflow was implemented. The expressed genes in sender cells−APCs were selected if they were expressed in at least 10% of that APC cell population. The gene set of interest in receiver cells−T cells was defined by adjusted $p$ value ≤ 0.05 and Log2FC ≥ 0.25 in the DEGs. Top 30 ligands that were further used to predict activated target genes and construct an activated ligand-receptor network. Default settings were used for all other parameters.

### Bulk RNA sequencing

Total RNA was extracted using RNeasy Plus Mini Kit (Cat# 74134, Qiagen) as per the manufacturer's instructions. RNA quantity check, preparation of RNA library, and mRNA sequencing were conducted by Novogene Co., LTD (CA, US). About 20 million paired-end 150 bp reads per sample were generated from Illumina NovaSeq 6000 Sequencing System. FASTQ raw reads were analyzed using the MTD pipeline[26]. Differential gene expression analysis between groups was done by DESeq2 R package (version 1.36.0)[75] with control for the subject effect. Genes with adjusted $p$ value < 0.05 were considered as differentially expressed. DEGs with $p$ values < 0.05 and |Log2FC|>0.5 were used for the IPA canonical pathway analysis. Different from single-cell (SC), to calculate the iAge index for bulk RNA-seq, normalized and transformed gene expression was multiplied with the gene's coefficient in the iAge gene set, then summed for each sample. Cellular senescence was scored using the ssGSEA[76] method on the SenMayo gene set. Cell Type Frequency Changes within PBMCs were predicted by CIBERSORTx Docker image−Fractions Mode version 1.0. Our single-cell RNA-seq data from PBMCs was used to build the Signature Matrix File as the reference to predict the cell proportion in the bulk RNA-seq data.

### Mouse spleen bulk RNA-seq analysis

Mouse spleen Bulk RNA-seq raw data was acquired from NASA GeneLab Data Repository with the accession ID: GLDS-420. Ten mice in space flight and ten mice in ground control (GC) were used in the experiment. The detailed study description and experiment protocols are on the data repository https://genelab-data.ndc.nasa.gov/genelab/accession/GLDS-420. MTD pipeline was used to process the FASTQ raw data, generate the count matrix, and then analyze differentially expressed genes between Flight and Ground groups.

### Gene set overlapping analysis

The $p$ value of gene overlapping between two datasets was calculated by Fisher's Exact Test in GeneOverlap R package [Shen L, Sinai ISoMaM (2022)]. *GeneOverlap: Test and visualize gene overlaps*. R package version 1.32.0 [http://shenlab-sinai.github.io/shenlab-sinai/]. The 375 DEGs in uG vs. 1G from unstimulated PBMCs single-cell RNA-seq results were used to match with the genes from PBMC bulk RNA-seq, I4, or JAXA studies. For the mouse genes in GLDS-420, we first convert them to the human orthologous before the analysis. In the matched genes, those expressions that were in the same log2FC direction as 375 DEGs as well as with $p$ value < 0.05, were considered overlapping (except for I4, where either direction was considered overlapping). Complete linkage hierarchical clustering was used to analyze dissimilarities in genes or pathways between datasets, and the results were visualized by the ComplexHeatmap R package (version 2.12.0)[77]. Moreover, the IPA canonical pathway analysis was performed on the matched genes of I4

and Twins studies. The 106 core gene set was constituted by DEGs that consistently change their log2FC directions in both SC and bulk data of PBMCs. The alteration of the core gene set by the compound was measured by Gene Set Enrichment Analysis (GSEA)[76,78] and Pearson correlation test.

## Compound analysis

FDA-approved drugs ($n = 1692$) are selected from the DrugBank database and food compounds ($n = 7962$) are selected from the FoodDB database as previously described[32]. LINCS compounds ($n = 5414$) are obtained from the LINCS L1000 project. 'Compound' is used as a general term for 'drug', 'food compound' and 'LINCS compound' throughout the document.

Compound-protein interactions are extracted from the STITCH database v5.0[79] by matching the InChI keys of drugs/food/LINCS compounds. STITCH collects information from multiple sources and individual scores from each source are combined into an overall confidence score. After processing, three datasets are obtained: (i) drug-gene interaction dataset containing 1890 drugs and 16,654 genes with 542,577 interactions (ii) food compound-gene interaction dataset containing 7654 compounds and 116,375 genes and 818,737 interactions (iii) LINCS compound-gene interaction dataset containing 5414 compounds and 16,794 genes and 692,152 interactions.

Statistical significance for the overlap between compound genes and the DEGs from the uG vs 1G of the unstimulated PBMCs single-cell RNA-seq is calculated using Fisher's Exact Test. The universal gene set contains all genes that interact with at least one compound. The compound with a low $p$ value interacts with a higher proportion of the DEGs than that expected by chance. Statistically significant compounds were then obtained after the Bonferroni adjustment of $p$ values. The pipeline for this compound analysis is implemented in the R script GCEA.

## Cell staining and imaging (super-resolution microscopy)

Live PBMCs were stained with 60 nM MitoTracker Red-CMX-Ros (ThermoFisher, Waltham, MA) either in 6-well plates or in the microgravity chambers for the last 2 hr of the microgravity simulation. At the end of the microgravity simulation cells were immediately fixed by 1:1 mixing the cell suspensions with 2× concentrated fixative (10% Sucrose (w/v) 120 mM KCl, 1% (w/v) glutaraldehyde, 8% (w/v) PFA pH 7.4) and incubated for 15 minutes at room temperature followed by 15 minutes on ice. Fixed cells were washed and stored in PBS until further staining for up to a week at 4 °C. 1 million fixed cells were resuspended in 1 mL of permeabilization solution (0.1% TritonX-100 in PBS) for 5 minutes. After twice washing in PBS, pellets were resuspended in 0.5 mL 1% BSA PBS containing Phalloidin-iFluor-488 (cat# ab176753, Abcam plc., Cambridge, UK) at the manufacturer's recommended dilution, and were incubated for 90 minutes with gentle agitation. After washing in PBS, cells were stained with Hoechst 33342 (1 μg/mL in PBS) for 10 minutes. The fixed-stained cells were immobilized at $3 \times 10^5$ cells per well density in glass-bottom 96-well microplates (Greiner Bio-One, Monroe, NC), which were pre-coated with polyethyleneimine (1:15,000 (w/v)) for 16 hours in a 37 °C incubator, and washed twice with PBS. Microplates with the cell suspensions were centrifuged in a swing plate rotor centrifuge (Eppendorf 5810 R) at $400 \times g$ and for 10 min and then fixed on the surface by adding an equal volume of 8% (w/v) PFA for 5 min. Finally, the fixative was replaced with 100 μL of antifade reagent (Vector Prolong Gold (ThermoFisher)). Samples were imaged immediately after this procedure.

## Image acquisition

Immobilized fixed-stained PBMCs were imaged on a Zeiss LSM980 Airyscan2 laser scanning confocal microscope (Carl Zeiss Microscopy, White Plains, NY). Single PBMCs were manually selected for recording based on low-resolution preview scans showing only nuclei. All singlet

cells were selected in a small neighborhood to avoid biases. In each microscopy session, 24-40 cells were selected for recording in one well for each condition. This was performed in an interleaved manner, capturing 6-8 cells at a time, and then moving to the next well and then repeating this multiple times using the Experiment Designer module for automation. Super-resolution volumes of ($358 \times 358 \times 70$ pixels, $0.035 \times 0.035 \times 0.13$ μm/voxel resolution) were recorded in the above-determined positions using Definite Focus autofocusing. A Plan-Apochromat $63 \times 1.40$ Oil lens, Airyscan2 SR (super-resolution) mode with optimal sampling and frame switching between 3 fluorescence channels to minimize spectral cross-bleed were used. MitoTracker Red, iFluor488, and Hoechs33342 were excited with 561, 488, and 405 nm solid-state lasers, respectively, using the optimal emission filter for each channel. 3D Airyscan2 processing was performed with standard filtering settings. With PBMCs from four donors, in six staining and microscopy sessions total of 930 valid volumes have been recorded, and are available at https://doi.org/10.5281/zenodo.8415196.

## 2-dimensional image analyses

Staining intensities, mitochondrial size, and punctate over diffuse index (defined as variance over mean) were determined in Image Analyst MKII 4.1.14 (Image Analyst Software, Novato, CA) in maximum intensity projection images using a custom pipeline available at https://github.com/gerencserlab/Superresolution-actin-and-mitochondria-analysis. Cellpose 2.0 with the "cyto2" neural network was used for finding cells in the images based on nuclear and actin staining[80]. Protruding actin bundles were analyzed by first binarizing projection images of actin using the trainable LABKIT segmentation[81], and this was followed by separation of protrusions and measurement of their maximal distance from the bulk of the cell using morphological erosion and distance image functions in Image Analyst MKII. Rescaled projection images were saved and further analyzed in CellProfiler 4.2.4[82], where images were segmented for nuclei and these segments were extended to the cell boundaries based on the phalloidin staining. These profiles were used for measuring shape, granularity spectrum, and texture in actin, mitochondria, and nuclei. For actin granularity spectrum measurement the following parameters were used in CellProfiler MeasureGranularity function: "Subsampling factor for granularity measurements" = 1, "Subsampling factor for background reduction = 0.125", "Radius of structuring element" = 12, "Range of granular spectrum" = 16. Similar results were obtained using a set of discrete Fourier transformation-based Butterworth bandpass filters in Image Analyst MKII for analysis of actin granularity spectrum changes in simulated microgravity and TLR stimulation. Here a series of 16 adjacent 4-pixels wide (in Fourier space of a $512 \times 512$ pixels image), 300-order bandpass filters with "Corrected Integral" normalization and absolute value calculation[83] were used starting at 1 pixel, and mean pixel intensities over whole cells in the filtered images were normalized to the unfiltered image. We have previously shown that this technique is primarily sensitive to sub-resolution changes in thickness of underlying filamentous structures[84], such as actin bundles in this case. We found no changes in granularity spectra measured by CellProfiler or Image Analyst MKII when analyzing mitochondria or nuclei of the same cells, excluding optical biases.

## 3-dimensional image analyses

Mitochondria:cell volume fraction was measured using a modification of the "Mitochondria:cell volume fractionator (basic)" pipeline in Image Analyst MKII[85], using the hole-filled actin image as cell marker and MitoTracker Red as mitochondrial marker, and all image planes to measure areas of mitochondrial and cell profiles. Cell and nucleus volumes and surface areas were measured using Imaris 9.9 (Oxford Instruments, Concord, MA) using the Cell and Batch modules.

For 2D and 3D image analyses, tabular data generated by Image Analyst MKII, CellProfiler, and Imaris were matched to conditions in

Microsoft Excel and in Mathematica 13 (Wolfram Research, Champaign, IL) and visualized in Prism 9 (GraphPad, La Jolla, CA) for statistical analysis. Two-tailed Welch's *t* test is used for all comparisons.

## Cdc42, Rac1, and RhoA G-LISA activation assay

PBMCs from different conditions were collected and $7 \times 10^6$ cells were lysed and snap-frozen immediately in liquid nitrogen. Cell lysate protein concentrations were measured using Precision Red Advanced Protein Assay Reagent (cat# ADV02, Cytoskeleton Inc., Denver, CO) and equalized. The GTP-bound Cdc42, Rac1, and RhoA levels were performed according to the manufacturer's protocol (cat# BK127-S, BK128-S, and BK124-S respectively, Cytoskeleton Inc.) and measured with a spectrophotometer at 490 nm.

## ROS detection

The abundance of ROS was measured via 2′,7′-dichlorodihydrofluorescein diacetate (DCFDA). Collected cells (100,000 cells per well) from each condition were incubated with 10 µM DCFDA Staining Buffer in dark at 37 °C for 30 minutes as per the manufacturer's suggestions (cat# 601520; Cayman Chemical, Ann Arbor, MI). The fluorescence was measured with a Pherastar FSx (BMG Labtech Inc., Cary, NC) microplate reader with the excitation wavelength at 495 nm, and emission at 530 nm.

## Luminex bead array

Cell culture media (supernatant) from all experimental conditions were separately collected and snap-frozen. Samples were sent to the Stanford Human Immune Monitoring Center and MILLIPLEX 48 Plex Premixed Magnetic Bead Panel (MilliporeSigma, Burlington, MA) was performed per the manufacturers' instructions.

## IFN ELISA level measurement

Cell culture media (supernatant) from microgravity and 1G ± R848 were separately collected at each experiment and snap-frozen. The samples were then thawed and used to detect the levels of IFNγ (cat# 430104; Biolegend Inc., San Diego, CA), IFNα all subtypes (cat# 41135; Pestka Biomedical Laboratories, Inc., Piscataway, NJ) per the manufacturers' instructions.

## IL ELISA level measurement

Cell culture media (supernatant) from microgravity and 1G were separately collected at each experiment and snap-frozen. The samples were then thawed and used to detect the levels of IL-8 (cat# 431504; Biolegend Inc., San Diego, CA), IL-6 (cat# 430504; Biolegend Inc., San Diego, CA), IL-1β (cat# 437004; Biolegend Inc., San Diego, CA) per the manufacturers' instructions.

## Flow cytometry

Single-cell suspensions from different donors and conditions were stained with LIVE/DEAD Fixable Blue Dead Cell Stain kit (cat# L34962; Invitrogen) for viability followed by Fc-blocking with human IgG (cat# AG714, Sigma-Aldrich) at room temperature for 10 mins. For staining of intracellular cytokines, single-cell suspension was stimulated with 1 µM R848 in the presence of 2.5ug/ml Brefeldin A (cat# 420601; Biolegend) for 9 hours prior to surface staining. The cells were further stained with fluorophore-conjugated surface antibodies for 20 min at 4 °C and intracellular antibodies for 30 min at room temperature following fixation and permeabilization using Foxp3 staining buffer set (cat# 00-5523-00; eBioscience). The surface and intracellular antibody panel are listed in Supplementary Data 19. Cell phenotyping was analyzed on a Cytek Aurora instrument and analyzed using FlowJo.

## JAXA transcriptomic data

JAXA cell-free RNA differential expression data was shared by Dr. Masafumi Muratani at the University of Tsukuba. Briefly, blood

samples were collected from six astronauts before, during, and after the spaceflight on the ISS. For this mission, we specifically made use of data from the samples of six astronauts, pooled into a single count, at day 5 and also at day 30, post-launch (i.e., in-flight), compared to pre-launch. In this study, human blood from astronauts was collected using Vacutainer EDTA-plasma separate gel collection tubes and centrifuged for 30 min at 3800 rpm ($1239 \times g$, ISS) or $1600 \times g$ (ground) before freezing at −95 °C (ISS) or −80 °C (ground). Cell-free RNA was purified from plasma samples through a TRIzol/chloroform method, sequenced (SMART-seq Stranded Kit, Takara Bio), and analyzed by the team leading the JAXA collaboration. Data was provided in csv format as normalized mean counts and normalized SEM of each gene at preflight and inflight time points. DEGs between 30 days in-flight and preflight time points were calculated by log2FC with *p* value < 0.05. The overlapping DEGs that are consistent in their log2FC directions with SC 375 gene signature are shown in Supplementary Fig. 16B.

## Inspiration4 mission data

Four astronauts' transcriptomic data from the Inspiration4 (I4) mission was collected by Dr. Christopher E. Mason and his team at Cornell University. Blood samples were collected before (preflight L-92, L-44, and L-3), during, and after (R + 1) the 3-day spaceflight in the SpaceX Dragon capsule. For this mission, we made use of data provided to us from the samples of the four astronauts comparing post-flight (R + 1) vs preflight (L-44) DEGs. Referring to the analysis workflow used by the I4 Cornell team, a list of fold change and *p* values based on post flight vs preflight findings was generated. The Seurat FindMarker parameters used to calculate I4 DEGs were the same as those used for our 375 DEGs. Next, the DEGs and pathway overlap from I4 single-cell analysis were calculated by using the methods described above in the section on Gene set overlapping analysis.

## NASA Twins study data

Gene expression data from the NASA Twins study[17] was provided by Cem Meydan in the Mason Lab in csv format and organized by Dr. Afshin Beheshti as normalized mean counts for four immune cell types. In brief, one astronaut was monitored before, during, and after a 1-year mission onboard the ISS, and his identical twin sibling was also monitored at the same time serving as a genetically matched ground control for this study. The NASA Twins study team provided DEGs in their four immune cell types with a list of fold change and *p* values based on inflight vs preflight findings. We reclustered our identified cell populations and computed DEGs accordingly. Next, the DEGs and pathway overlap from the Twins analysis were calculated by using the methods described above in the section on Gene set overlapping analysis.

## ROS reduction compound

Quercetin (Sigma-Aldrich, St Louis, MO) stock solution was prepared with DMSO at 1000×. In the cell culture experiments utilizing quercetin, the concentration of quercetin was decided based on existing literature[86–89]. After incubation with quercetin, cells were counted with a Cellometer Auto 2000 Cell Viability Counter (Nexcelom, San Diego, CA), which utilizes Acridine Orange and Propidium iodide dual-staining systems to accurately distinguish live vs dead cells. After 25 hours of 50 µM quercetin treatment, the cell viability across PBMCs in both 1G and simulated microgravity conditions were at least 93%. There were no statistical differences in viability observed between the groups with and without quercetin treatment.

## Statistical analyses

In addition to the methods described above, the Wilcoxon Rank Sum Test was used to assess whether the distributions of data from cell score or microbial abundance were significantly different between the 1G and uG cell populations from single-cell data. The association

between single-cell and bulk RNA-seq in gene expressions was tested by Spearman's correlation. Mann–Whitney test was performed on ROS reduction by quercetin. Unpaired parametric two-tailed $t$ tests were performed on single-cell iAge, SenMayo, and imaging analyses for statistics. G-LISA, ELISA, Luminex, and DCFDA results were assessed by paired parametric two-tailed $t$ test. However, given our existing transcriptomic and cytokine data showed decreased interferon coupled to increased IL-1β, IL-6, and IL-8 production in microgravity, for validation the flow cytometry results were assessed by one-tail paired parametric $t$ test. In PBMCs bulk RNA-seq results, the difference in iAge and Sen-Mayo scores of samples with or without compound treatment was evaluated by two-tailed paired parametric $t$ test. R (version 4.2.0) and GraphPad Prism 9 were used to conduct the statistical analyses. Significance was set at 0.05. Outliers in datasets were assessed using Grubbs' test (alpha = 0.01) and specified in figure legends if any were removed.

### Reporting summary

Further information on research design is available in the Nature Portfolio Reporting Summary linked to this article.

## Data availability

The raw single-cell and bulk RNA-seq data generated in this study are deposited in the Gene Expression Omnibus (GEO) database under accession code GSE218937. Mouse spleen Bulk RNA-seq raw data was acquired from NASA GeneLab Data Repository with the accession ID: GLDS-420. All other data are available in the article and its Supplementary files or from the corresponding author upon request. Source data including cell proportion, gene set score, GLISA, ELISA, Luminex, and flow results are provided as a Source Data file. Source data are provided with this paper.

## Code availability

The code used for the analysis of sequencing data is available at GitHub and Zenodo repository [https://github.com/FEI38750/Immune_Dysfunction_in_Microgravity; https://zenodo.org/record/8247816][90]. Code for GCEA is available upon request from the corresponding author.

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

## Acknowledgements

We would like to thank the crew members of I4, JAXA, and Twins study for contributing valuable samples to improve our understanding of the immune system in space. We would like to thank Dr. Cheryl Nickerson (Arizona State University) for technical discussions on simulated microgravity ground analogs, and Dr. Brian Crucian (NASA) for discussions on translational aspects of the immune system in space. We would like to thank Dr. Christopher Chin (Cornell University), Dr. Remi V. Klotz (University of Maryland) and Dr. Min Yu (University of Maryland) for assistance with I4 single cell preparation and analysis. We would like to thank Ryan Kwok and Ritesh Tiwari for technical assistance with flow cytometry. This work was supported in part through funds derived from the Buck Institute for Research on Aging (D.A.W., D.F.), and the Huiying Memorial Foundation (D.A.W.). T.R.V. and J.B. are funded by a T32 NIH fellowship grant (NIA T32 AG000266). C.E.M. thanks the Scientific Computing Unit (SCU) at WCM, the WorldQuant Foundation, NASA (NNX14AH50G, NNX17AB26G, 80NSSC22K0254, NNH18ZTT001N-FG2, 80NSSC22K0254, NNX16AO69A), the National Institutes of Health (RO1MH117406), and LLS (MCL7001-18, LLS 9238-16). Figure 7A was created with BioRender.com.

## Author contributions

F.W., H.D., D.F., and D.A.W. contributed to the experimental design, execution, and analysis of the study. F.W., H.D., and D.A.W. drafted the manuscript, and D.F. and S.W. edited the manuscript. H.D., F.W., C.A.L., N.M., J.M.B., H.H. P.M., T.R.V., H.G.K., N.S., and A.A.G. assisted with in vitro and experiment execution and analysis of data. F.W., H.D., M.F., J.B., and K.N. performed computational analyses of single-cell datasets or validation data. E.O., C.M., J.K., and C.E.M. led I4 data acquisition and sharing. M.M. led JAXA study data acquisition and sharing. A.A.G. performed microscopy image analysis. C.M., C.E.M., F.G-B., and A.B. led Twins study data acquisition and sharing. S.M., M.M., A.A.G., A.B., and C.E.M. supervised aspects of the study. D.F. and D.A.W. supervised the overall implementation of the study. All authors had an opportunity to view and edit the manuscript.

## Competing interests

D.F. is a founder of Edifice Health, a company that utilizes inflammatory biomarkers (e.g., iAge) to predict age-related diseases. D.F., C.E.M., and D.A.W. are co-founders of Cosmica Biosciences, a company that studies altered biological aging in spaceflight exposures. J.B. and F.W. are stakeholders in Cosmica Biosciences. A.A.G. has a financial interest in Image Analyst Software, which makes the software Image Analyst MKII used here for image analysis. The Buck Institute (F.W., H.D., J.B., D.F., D.A.W.) has filed a patent application (application number: 63/520,327) on using simulated microgravity to mimic and counter aging and spaceflight effects in cells based on data from this manuscript. The remaining authors declare no competing interests.

## Additional information

[1]Buck Institute for Research on Aging, Novato, CA 94945, USA. [2]Leonard Davis School of Gerontology, University of Southern California, Los Angeles, CA 90089, USA. [3]Department of Physiology and Biophysics, Weill Cornell Medicine, New York, NY 10021, USA. [4]Department of Immunology, University of Toronto, Toronto, ON M5S 1A8, Canada. [5]Department of Laboratory Medicine and Pathobiology, University of Toronto, Toronto, ON M5S 1A8, Canada. [6]Pathology and Laboratory Medicine, Mount Sinai Hospital, Toronto, ON, Canada. [7]Department of Medicine, University of Virginia, Charlottesville, VA, USA. [8]Department of Biochemistry and Molecular Genetics, University of Virginia, Charlottesville, VA, USA. [9]Stanford Cardiovascular Institute, Stanford University School of Medicine, Stanford, CA 94305, USA. [10]Transborder Medical Research Center, University of Tsukuba, Ibaraki 305-8575, Japan. [11]Department of Genome Biology, Faculty of Medicine, University of Tsukuba, Ibaraki 305-8575, Japan. [12]Blue Marble Space Institute of Science, Space Biosciences Division, NASA Ames Research Center, Moffett Field, CA 94043, USA. [13]Stanley Center for Psychiatric Research, Broad Institute of MIT and Harvard, Cambridge, MA 02142, USA. [14]The HRH Prince Alwaleed Bin Talal Bin Abdulaziz Alsaud Institute for Computational Biomedicine, Weill Cornell Medicine, New York, NY 10021, USA. [15]WorldQuant Initiative for Quantitative Prediction, Weill Cornell Medicine, New York, NY 10021, USA. [16]The Feil Family Brain and Mind Research Institute, Weill Cornell Medicine, New York, NY 10021, USA. [17]Stanford 1000 Immunomes Project, Stanford University School of Medicine, Stanford, CA, USA. [18]Institute for Research in Translational Medicine, Universidad Austral, CONICET, Pilar, Buenos Aires, Argentina. [19]Division of Cellular & Molecular Biology, Toronto General Hospital Research Institute (TGHRI), University Health Network, Toronto, ON M5G 1L7, Canada. [20]These authors contributed equally: Fei Wu, Huixun Du. [21]These authors jointly supervised this work: Christopher E. Mason, David Furman, Daniel A. Winer. ✉e-mail: chm2042@med.cornell.edu; DFurman@buckinstitute.org; dwiner@buckinstitute.org

