## [Peer Review File · Nature Communications]

Single Cell Analysis Identifies Conserved Features of Immune Dysfunction in Simulated Microgravity and SpaceflightREVIEWER COMMENTS

Reviewer #1 (Remarks to the Author):

This study examines the impact of microgravity on the transcriptomic profiles of peripheral immune cells exposed to 25 hours of simulated microgravity using single-cell RNA sequencing. Samples were examined under microgravity alone as well as in microgravity following stimulation with R848. These results demonstrate that microgravity altered the immune landscape, particularly in monocytes. Interestingly, the study found that microgravity activated expression of endogenous retroviral and mycobacterial transcripts. The authors complete their study by examining possible therapeutic interventions for microgravity-induced changes in immune function. These in vitro results were validated against data from multiple low Earth orbit missions. Overall, this is an interesting study conducted with state-of-the-art single-cell sequencing methods and a rigorous approach, including multiple validation experiments. The results are noteworthy and will be of interest to the fields of aviation and space physiology. There are some missing methodological details and discussion points which may improve the manuscript. In general, a description of the physiological principals linking microgravity exposure to changes in immune cell shear stress and resulting cell phenotypes in vivo would greatly strengthen the paper.

I provide specific comments and questions below.

Methods:

- Were blood samples collected while donors were fasting?
- Were hemoglobin transcripts removed before RNA library prep?
- When analyzing the bulk RNA seq gene expression changes in response to microgravity, did the statistical model include corrections for changes in cell distributions? If not, the changes in gene expression could be driven primarily by changes in immune cell subpopulations.
- General comment: It would be interesting to see how other stimulating factors were impacted by microgravity (i.e., LPS, etc.), but this is not a major concern.
- In the in vitro studies, comparisons are made across 1G vs. uG groups, as well as 1G stimulated vs. uG stimulated groups. It would be interesting to compare expression of key inflammatory genes across all 4 groups simultaneously to demonstrate if uG either blunts or

exacerbates the inflammatory response to the TLR7/8 agonist.

-There is a minimal description of the experiments using DrugBank, LINCS etc. and treatments with quercetin in the main text methods section. How was the treatment concentration determined and optimized? Was cell viability after treatment measured?

Results and Figures:

-Figure 1B: It may not be necessary to show the changes in immune cell subpopulations with so many different visuals. The log fold changes should be sufficient. It is also difficult to see the differences in populations based on color in the lower left panel. The colors are difficult to tell apart.

-In the third paragraph of the Results section, it is stated that there are decreases in RAC, FAK, HIF-1 signaling, acute phase response signaling and oxidative stress signaling based on the IPA analysis. However, Figure 1F does not seem to visually support this. For example, HIF-1 signaling increased in some cell populations and decreased in at least 3 others.

-It seems that TLR7/8 stimulation in microgravity in this study lead to increased CD14 monocytes at the expense of reduced NK cells (if I'm reading the colors correctly in Figure 2B lower left panel) – this is very interesting. However, the text describing this figure (third paragraph on page 6) states that there was also reduced expansion of CD4+T cells and CD8 T effector memory cells, but this doesn't seem to be the case based on Figure 2B (CD4 TCM went from 58.4% to 54.4%). Again, I may be mismatching the colors from the plot and legend because the colors are very similar.

-It would be helpful in the legend for figure 3 to clarify if/when the comparisons for the i4 group are changes in gene expression post-flight versus pre-flight. Also, state again here the description for the "SC" abbreviation. My understanding is that these are genes differentially expressed via single cell sequencing of PBMCs cultured in microgravity versus 1G.

-Figure 4B is a bit difficult to read. Ease of matching on x and y axes may be improved by adding lines to delineate columns and rows.

-In figure 4C, is the Q_treatment group showing results for uG versus 1G with both groups receiving quercetin treatment? Or was the quercetin treatment only administered to the uG group in that comparison?

-Since quercetin treatment reduced expression of 60% of genes, it would be useful to

include data on cell viability and metabolic activity following quercetin treatment using a live/dead stain and/or ATP production assay (such as CellTiter-Glo or comparable assay).

Was cell viability unaffected? Was cell metabolic activity unaffected?

-How was the concentration of quercetin treatment determined and were concentration optimization assays performed?

Discussion:

-To what extent can the in vivo inflammatory responses and latent virus reactivation be explained by overall stress of spaceflight versus microgravity? Psychological and physiological stress have also been independently associated with viral reactivation. This would be an interesting discussion point.

-The data suggest that (at least in in vitro cultures) microgravity may change cytoskeletal components and that this may be due to changes in mechanical stress on the cells, which would also contribute to transcriptional changes and inflammatory status. This is very interesting, however, there is no discussion of how this hypothesis translates in vivo. How would mechanical stress and strain on these cells change in vivo when an individual experiences microgravity? Central pooling of blood will occur in microgravity, which may have some impact on mechanical forces on the cells, but they would still experience pulsatile and intermittent shear forces. Can the authors discuss their thoughts on these in vivo mechanisms? This would be a key discussion point to explain how microgravity may cause these transcriptional and cytoskeletal changes in vivo.

Reviewer #2 (Remarks to the Author):

The paper by Wu et al investigates the effects of simulated microgravity on immune cell distribution, immune cell transcriptome at the scRNAseq and bulk RNAseq levels, and innate response to TLR ligands. They further compare their data with existing spaceflight data sets from humans and mice. These investigations identify potentially interesting changes in the immune system in response to microgravity. Finally, the authors identify the plant flavonoid antioxidant Quercetin as a potential treatment to inhibit some effects on the immune system in microgravity. The findings are interesting and provocative. This reviewer nevertheless has a few concerns:

- 1) It would be very good to confirm some of the findings with alternative methodologies. For example, the changes in immune cell subset representation in response to simulated microgravity shown in Figure 1 B would be much strengthened by confirmatory experiments using flow cytometry.
- 2) Flow cytometry could also be used to assess if the altered cell distribution pattern derives from a differential susceptibility to apoptosis.
- 3) Flow cytometry could also be used to assess if the simulated microgravity alters the function of immune cells.
- 4) It is unclear what conclusions the authors draw from the trajectory data presented in Figure 1E.
- 5) The meta transcriptional data on virus reactivation in Figure 1I is very superficially described. How was this done and what conclusions can be drawn? Can a similar signal be seen in the confirmatory data sets available?
- 6) The title emphasizes “dysfunction” of the immune system in microgravity and spaceflight. Was dysfunction really shown, or is “changes” a better way to describe the changes?
- 7) Overall, the manuscript has a somewhat preliminary feel and the writing could be more direct and to the point.

Reviewer #3 (Remarks to the Author):

This is a novel study that applies single cell genomics to study the association of microgravity with immunological dysfunction. They characterize altered genes and pathways across immune cells under basal and stimulated states with a Toll like Receptor-7/8 agonist. At basal state, simulated microgravity altered the transcriptional landscape across immune cells, with monocyte subsets showing most pathway changes. Results from single cell analysis were validated by RNA sequencing and super-resolution microscopy. They provide evidence that microgravity has an impact on pathways essential for optimal immunity. Finally, they used machine learning to identify compounds that can reverse abnormal pathways induced by microgravity. The paper reads interesting. It should be considered for publication. However, I have only one comment on the single cell experiment design.

“unstimulated PBMCs single-cell transcriptomes (10X Genomics), pooled together from a male (36 years old) and a female (25 years old) donor, that underwent either 1G or simulated microgravity (uG) for 25 hours total.” Lack of replicates is a big problem for the single cell experiment. However, the authors are very lucky that they pooled PBMC from one male and one female. It should be fairly easy to differentiate male single cells from female single cells using the RNA-seq raw data (by Y chromosome gene or SNP). In this way, the authors should be able add statistical power to their results. It should also be possible then to evaluate the batch effect and male female difference in their experiment. Lots of analyses need to be revised after separating male data from the female data.

REVIEWER COMMENTS

Reviewer #1 (Remarks to the Author):

This study examines the impact of microgravity on the transcriptomic profiles of peripheral immune cells exposed to 25 hours of simulated microgravity using single-cell RNA sequencing. Samples were examined under microgravity alone as well as in microgravity following stimulation with R848. These results demonstrate that microgravity altered the immune landscape, particularly in monocytes. Interestingly, the study found that microgravity activated expression of endogenous retroviral and mycobacterial transcripts. The authors complete their study by examining possible therapeutic interventions for microgravity-induced changes in immune function. These in vitro results were validated against data from multiple low Earth orbit missions. Overall, this is an interesting study conducted with state-of-the-art single-cell sequencing methods and a rigorous approach, including multiple validation experiments. The results are noteworthy and will be of interest to the fields of aviation and space physiology. There are some missing methodological details and discussion points which may improve the manuscript. In general, a description of the physiological principals linking microgravity exposure to changes in immune cell shear stress and resulting cell phenotypes in vivo would greatly strengthen the paper.

I provide specific comments and questions below.

Methods:

-Were blood samples collected while donors were fasting?

The donors were not fasted. Blood samples were collected at the Stanford blood center, where they drew a standard quantity of blood for transfusion. This blood was processed into buffy coats by the blood center. Since the remaining non-buffy components go for blood transfusion, the donors are typically asked to be non-fasted. This point is now clarified in the text.

-Were hemoglobin transcripts removed before RNA library prep?

Thank you for your query regarding the potential presence of hemoglobin transcripts in our RNA library preparation. In our protocol, we did not employ a library preparation kit designed to remove hemoglobin transcripts. This decision was based on the nature of our samples, which were peripheral blood mononuclear cells (PBMCs) isolated from human buffy coats.

PBMCs do not typically produce hemoglobin, a molecule primarily synthesized by red blood cells (RBCs), though they may contain receptors and proteins linked to iron handling and metabolism. While our samples may still contain a very small fraction of RBCs, we addressed this issue by filtering out these cells during the single-cell RNA-seq bioinformatic analysis process. We have updated the text to reflect this protocol – thank you. Furthermore, our sequencing depth (> 20 million reads/sample in Bulk RNAseq and > 300 million reads/sample in Single-cell RNAseq) is sufficient to detect and quantify non-hemoglobin transcripts of interest.

In light of a very low RBC content in our PBMC sample due to filtering, and the sequencing depth, we considered the influence of hemoglobin transcripts on our data to be minimal to negligible. To support this notion, we have now quantified the percentage of hemoglobin transcript¹ reads in the total trimmed

reads in our samples. These percentages range from 0.02% to 0.73%, as illustrated in the accompanying image.

For these reasons, while the potential presence of hemoglobin transcripts was considered, our sample preparation and analysis methods effectively mitigated any substantial impact on our data. Moreover, all samples were processed in an analogous manner, also limiting any confounder effects of hemoglobin transcripts from RBCs from one sample to another.

-When analyzing the bulk RNA seq gene expression changes in response to microgravity, did the statistical model include corrections for changes in cell distributions? If not, the changes in gene expression could be driven primarily by changes in immune cell subpopulations.

The bulk RNA-seq analysis did not correct the changes in cell distributions. It is because the relationship between gene expression and cell type proportions in PBMCs can be complex and not always linear. Gene expression levels in different cell types can vary due to various factors, such as cell type-specific functions, activation states, or subpopulations. Additionally, although we can predict the cell type proportion by using the deconvolution tool (e.g., CIBERSORTx), we cannot get the gene expression split among those cell types. Because many genes are expressed in a cell type-specific manner, therefore, when correcting for cell type proportions in bulk RNA-seq data by including cell type proportions as covariates in a model (e.g., DESeq2), misleading results may be generated due to non-linearities in the relationship between gene expression and cell proportions. Nonetheless, to better limit the potential confounder of varying cell distributions between people and conditions in the bulk RNA seq validation cohort, we have doubled the size of this cohort from three to six samples in the current submission to increase rigor (Figure 3A-C). We also performed CIBERSORTx analysis on this larger cohort to specifically illustrate cell proportion differences between donors and conditions, and included this data in the new supplemental figure 15.

-General comment: It would be interesting to see how other stimulating factors were impacted by microgravity (i.e., LPS, etc.), but this is not a major concern.

-In the in vitro studies, comparisons are made across 1G vs. uG groups, as well as 1G stimulated vs. uG stimulated groups. It would be interesting to compare expression of key inflammatory genes across all 4 groups simultaneously to demonstrate if uG either blunts or exacerbates the inflammatory response to the TLR7/8 agonist.

Thank you for giving these suggestions. To better understand if uG blunts or exacerbates the inflammatory response to TLR7/8 agonist, we have added in a large volume of data which is displayed as two new supplementary figures (Supplementary Figure 12 and 13). The single-cell data was analyzed for both 1G stimulated vs 1G unstimulated, and uG stimulated vs uG unstimulated shown in Supplementary Figure 12. Next, to determine the sensitivity of individual immune cell populations to TLR7/8 agonist in 1G vs simulated microgravity, we compared the differences in responsiveness to stimulation. We subtracted the fold change induction in 1G from induction in simulated microgravity to determine sensitivity to stimulation. This data is shown in Supplementary Figure 13A. Interestingly, across "Overall" category of immune cells, we see a pattern of reduced responsiveness to TLR7/8 stimulation in simulated uG to most of the highest genes induced at 1G, with T cells and NK cells showing some of the most blunted inflammatory gene induction. Some of these genes included those in interferon signaling, and GBP genes. Interestingly, monocytes tended to maintain such responses better in simulated uG, consistent with their predisposition to some inflammatory pathways in simulated uG.

For further comparisons, we also mapped out the top 50 conserved DEGs specifically sensitive to simulated uG across immune populations, regardless of stimulation condition. This data is shown in Supplementary Figures 13B-D. We have included the description of these new comparison results in the result section "Simulated Microgravity Alters the Transcriptional Landscape of Individual Immune Cells", as well as in the methods.

Thank you for the suggestion of LPS and other TLR agonists. We are also interested in pursuing this work as a follow up study to the current paper.

-There is a minimal description of the experiments using DrugBank, LINCS etc. and treatments with quercetin in the main text methods section. How was the treatment concentration determined and optimized? Was cell viability after treatment measured?

The concentration of quercetin is decided based on existing literature²⁻⁵. Cells were counted with Cellometer Auto 2000 Cell Viability Counter from Nexcelom. This machine utilizes Acridine Orange and Propidium iodide dual-staining systems to accurately distinguish live vs dead cells. This information is now added to the methods. After 25 hours of 50uM quercetin treatment, we do not see significant reduction in cell viability across PBMCs. See figure below.

Results and Figures:

-Figure 1B: It may not be necessary to show the changes in immune cell subpopulations with so many different visuals. The log fold changes should be sufficient. It is also difficult to see the differences in populations based on color in the lower left panel. The colors are difficult to tell apart.

Thank you for your feedback. We agree with your assessment regarding the depiction of immune cell subpopulation changes. To address your concerns, we have made the following modifications to the previous Figure1b and 2b.

First, we have optimized the presentation of changes in immune cell subpopulations, choosing to primarily focus on log fold changes. Next, we have eliminated the stacked bar plot from our figures, as it seems to contribute less to the overall interpretation of the data. Finally, we have enlarged the remaining bar plots in Figures 1B and 2B to enhance their clarity and readability. We hope these changes make our findings more comprehensible and easier to interpret.

-In the third paragraph of the Results section, it is stated that there are decreases in RAC, FAK, HIF-1 signaling, acute phase response signaling and oxidative stress signaling based on the IPA analysis. However, Figure 1F does not seem to visually support this. For example, HIF-1 signaling increased in some cell populations and decreased in at least 3 others.

We thank you for this observation and have updated the text to better reflect Figure 1F. Pathways pertaining to RAC, FAK, HIF-1 signaling, acute phase response signaling and oxidative stress signaling are indeed increased in the “Overall” category of immune cells and the text now reflects this observation. On a side note, we would like to clarify that the “overall” immune cell signaling pathway is driven in part by the transcript of the most abundant cell type. Consequently, it is anticipated that the overall direction of signaling enrichment may not perfectly correlate with the direction of the pathways observed in every cell subtype. This phenomenon could account for the variation seen in HIF-1 signaling among different cell populations. These variations of course are one reason why we validated the single-cell data across multiple additional datasets, some of which are shown in Figure 4E.

-It seems that TLR7/8 stimulation in microgravity in this study lead to increased CD14 monocytes at the expense of reduced NK cells (if I’m reading the colors correctly in Figure 2B lower left panel) – this is very interesting. However, the text describing this figure (third paragraph on page 6) states that there was also reduced expansion of CD4+T cells and CD8 T effector memory cells, but this doesn’t seem to be the case based on Figure 2B (CD4 TCM went from 58.4% to 54.4%). Again, I may be mismatching the colors from the plot and legend because the colors are very similar.

We are sorry for the confusion. The description in the manuscript now has been modified based on these suggestions to remove mentioning of CD4 T cells.

-It would be helpful in the legend for figure 3 to clarify if/when the comparisons for the i4 group are changes in gene expression post-flight versus pre-flight. Also, state again here the description for the “SC” abbreviation. My understanding is that these are genes differentially expressed via single cell sequencing of PBMCs cultured in microgravity versus 1G.

Thank you for the suggestion. For Fig3F and 3G, i4 the comparison is post-flight (R+1 = 1 day after Return to ground) vs pre-flight (L-44 = 44 days before launch). And I4 is compared with single-cell results, and to make this clearer we added the full name on the figure legend. Note as well the improved analysis of the i4 dataset. In the previous submission, we were only provided i4 gene expression data from our core list of DEGs in our single cell data. This new Figure 3F and 3G now compares our single-cell DEG signature against the entire dataset from the i4 mission for improved completeness. Most of the core pathways from our previous submission remain highlighted in the current more complete analysis. These changes are also reflected in the results and methods sections.

-Figure 4B is a bit difficult to read. Ease of matching on x and y axes may be improved by adding lines to delineate columns and rows.

Thank you for your feedback regarding Figure 4B. We understand your suggestion to add lines to delineate columns and rows to improve the matching on the x and y axes. However, after careful consideration and several trials, we have decided to maintain the current format. We found that the addition of such lines, while potentially aiding in the alignment of the axes, could inadvertently clutter the figure, making it harder to read and interpret. We believe that the current format strikes a balance

between clarity and simplicity, and it allows the reader to focus on the data itself without being distracted by extraneous gridlines. We hope you understand our rationale for retaining the current graphical representation.

-In figure 4C, is the Q_ treatment group showing results for uG versus 1G with both groups receiving quercetin treatment? Or was the quercetin treatment only administered to the uG group in that comparison?

Sorry for the confusion on figure 4C. Quercetin was only administered to uG group. The comparison is between the "uG Q- vs 1G Q-" (uG vs 1G) and "uG Q+ vs 1G Q-" (uG+Q vs 1G). We have revised the figure labels to ensure this distinction is clear. We chose these conditions for comparison since they are both relative to the same 1G with no Q control.

-Since quercetin treatment reduced expression of 60% of genes, it would be useful to include data on cell viability and metabolic activity following quercetin treatment using a live/dead stain and/or ATP production assay (such as CellTiter-Glo or comparable assay). Was cell viability unaffected? Was cell metabolic activity unaffected?

Thank you for the comment. To clarify, quercetin did not just reduce expression of 60% of the genes, it reversed the directionality of these genes, including many genes that become induced. These results are captured in the revised figure 4C, which also includes the comparison to a new list of 106 overlapping genes based on a higher n value of samples. This new analysis now shows a reversal rate of 70% (74/106 genes), improved over the previous version. Related to this data, there was no difference in cell viability in the samples assessed, as explained earlier based on our cell counts using the Cellometer Auto 2000 Cell Viability Counter. This information is now added to the methods of the paper. In terms of metabolic activity, we improved on our ROS measurement assay (one output that can be linked sometimes to mitochondrial dysfunction), repeating the experiment with true biological replicates instead of the previous technical replicates in the old version of the paper. This new data validated our previous findings using the technical replicate samples. Effects of quercetin on metabolism has been previously reported in cells, including on immune cells and on ATP production. For instance, based on previous literature, quercetin can reprogram metabolism and increase ATP production in macrophages: ⁶PMID: 36982615. Quercetin is a strong antioxidant and anti-inflammatory compound that can inhibit inflammatory cytokine production in various immune cell populations⁷. Quercetin has been found to improve mitochondrial function in many non-immune cells⁸⁻¹⁰. These data are supported by our more rigorous new data on quercetin reducing uG associated ROS. Given the multiple existing literature on this topic and due to the strict timelines for resubmission, we decided not to work up additional metabolic parameters beyond improving our ROS data post quercetin. However, we do agree that this topic, and the effects of uG on cell metabolism in general, is good for a new paper, which we plan to work on next.

-How was the concentration of quercetin treatment determined and were concentration optimization assays performed?

As previously described, we determined the concentration of quercetin based on existing literature ²⁻⁵.

Discussion:

-To what extent can the in vivo inflammatory responses and latent virus reactivation be explained by overall stress of spaceflight versus microgravity? Psychological and physiological stress have also been independently associated with viral reactivation. This would be an interesting discussion point.

Thank you for the suggestion. We have added in psychological and physiological stressors to our overall model of spaceflight impact on the immune system in the discussion of the paper, including potential mechanisms also contributing to viral reactivation in space.

-The data suggest that (at least in in vitro cultures) microgravity may change cytoskeletal components and that this may be due to changes in mechanical stress on the cells, which would also contribute to transcriptional changes and inflammatory status. This is very interesting, however, there is no discussion of how this hypothesis translates in vivo. How would mechanical stress and strain on these cells change in vivo when an individual experiences microgravity? Central pooling of blood will occur in microgravity, which may have some impact on mechanical forces on the cells, but they would still experience pulsatile and intermittent shear forces. Can the authors discuss their thoughts on these in vivo mechanisms? This would be a key discussion point to explain how microgravity may cause these transcriptional and cytoskeletal changes in vivo.

The reviewer raises interesting points here. We believe that the combination of microgravity induced innate cytokine and inflammatory gene action, works alongside of changes to shear forces and increased pressures in some tissues associated with spaceflight. Both increased pressures within tissues and simulated microgravity can induce cytokines such as IL-6, or tune the immune system to respond more robustly to TLR agonist and make cytokines like IL-1b¹¹. Multiple mechanosensing pathways and mediators including Taz/Yap (Hippo signaling), MRTFA, Piezo1, Trpv4, and the SUN complex are altered to changing mechanical forces within immune cells. While some of these pathways have been put forward as mediators of microgravity effects in cells¹²⁻¹⁴, more work is needed to dissect out the signaling events. We have added a few lines to the discussion supporting this avenue of future research. Thank you.

Reviewer #2 (Remarks to the Author):

The paper by Wu et al investigates the effects of simulated microgravity on immune cell distribution, immune cell transcriptome at the scRNAseq and bulk RNAseq levels, and innate response to TLR ligands. They further compare their data with existing spaceflight data sets from humans and mice. These investigations identify potentially interesting changes in the immune system in response to microgravity. Finally, the authors identify the plant flavonoid antioxidant Quercetin as a potential treatment to inhibit some effects on the immune system in microgravity. The findings are interesting and provocative. This reviewer nevertheless has a few concerns:

1) It would be very good to confirm some of the findings with alternative methodologies. For example, the changes in immune cell subset representation in response to simulated microgravity shown in Figure 1 B would be much strengthened by confirmatory experiments using flow cytometry. 2) Flow cytometry could also be used to assess if the altered cell distribution pattern derives from a differential susceptibility to apoptosis. 3) Flow cytometry could also be used to assess if the simulated microgravity alters the function of immune cells.

Thank you for these suggestions. Since our changes to cell proportions in the unstimulated state shown in figure 1B were relatively small, and a previous paper¹⁵ had already performed flow/mass cytometry on unstimulated cells after 22 hours of simulated microgravity and found no major differences in their n=8 samples, we decided to focus our flow cytometry on the stimulated samples with the intent of assessing function. We updated the main body text to reflect the relatively small changes to cell proportions in the unstimulated states. However, it is important also to note that flow cytometry assays may not be the ideal way to assess changes directly occurring in simulated microgravity through use of surface markers. While our transcriptome data captures transcripts from cells immediately upon removal from the microgravity chamber, flow cytometry captures surface marker changes about an hour post simulated microgravity exposure when the cells are fixed. The delay in flow cytometry occurs while the cells are processed under 1G conditions for surface and intracellular staining, which does present a confounding variable. This confounding variable on how cells respond to the return to gravity may add variability to the flow data. However, for stimulation data, we were able to add in Brefeldin A directly to the cells during microgravity exposure to capture altered cytokine production during simulated microgravity, which is another reason we decided to focus on stimulation data only by flow cytometry.

Thus, we took multiple approaches to improve on our cell quantification and functional assessments at the advice of the reviewer. First to better assess changes to cell proportions with no stimulation (ie: simulated microgravity alone vs 1G), we doubled the size of our bulk RNA seq cohort from n=3 to n=6, which captures transcripts immediately from simulated microgravity, and then performed CIBERSORTx algorithms to predict cell types and frequencies in the sample. On our n= 6 donors, we found mostly small changes in similar directions to the single cell data with small reductions in B intermediate cells and NK cells, coupled to increases in CD14 monocytes, and reductions in CD16 monocytes. T cells were a bit more variable as CD8 T cells followed similar trending directions as in single cell data, while CD4 subsets were sometimes in the reverse direction. This work was added as a new Supplemental Figure 15.

To first validate the functional impact of simulated microgravity on cumulative immune cell function (especially pertaining to inflammatory function) we performed a 48 plex human Luminex assay on n=12 donors, measuring 48 different cytokines and secreted markers, with targeted ELISA validation on both unstimulated and stimulated PBMCs after 25 hours of culture (16 hour conditioning + 9 hours TLR7/8 stimulation, or 25 hours in the unstimulated case) in both 1G and simulated microgravity conditions. This data was used to create a new figure, Supplementary Figure 19. Again, as the supernatants are harvested immediately from the simulated microgravity chamber, this assessment captures the functional capacity of the cells during simulated microgravity exposure alone, with no extra time under

the return to 1G (which is captured by flow cytometry during processing prior to fixation). Interestingly, consistent with both our single cell and bulk RNA sequencing data, simulated microgravity was associated with increased or trending increases in mainly innate/monocyte immune cell derived inflammatory cytokines and chemokines (e.g. IL-6, IL-8, IL-12p40, CCL4), coupled to reduction in cytokines that associate with T cell activation, or proliferation (e.g. IL-2, IL-7, IL-15). Upon stimulation with TLR7/8 agonist, simulated microgravity further facilitated significant (IL-1 β) or trending (IL-8) increases by Luminex, coupled to significant or trending reductions in IL-2, IFN γ and IFN α .

To further assess functional changes to immune cells in simulated microgravity vs 1G conditions in response to TLR7/8 stimulation, we used intracellular flow cytometry (new Supplementary Figure 20 and 21) for core cytokines IL-1 β , IL-6 and IFN γ frequently altered by Luminex and ELISA. We saw increased IL-1 β production across all characterized monocyte populations in simulated microgravity, and variably increased IL-6 only in some cell populations. NK cells also showed a reduction in the proportion producing IFN γ , as well as reduced proportions expression activation marker, CD69, and degranulation marker, LAMP-1, consistent with reduced functionality and response to stimulation in simulated microgravity. T cell subsets were less altered, though we still detected near significant or significant reductions in the proportions of CD4+ and CD8+ central memory T cells expressing activation marker, CD69 (consistent with previous work, ¹⁵PMID: 34099760, using a different form of stimulation), and in effector memory CD4+ T cells expressing proliferation marker, Ki67.

Overall, these new findings from flow cytometry, Luminex, and/or ELISA demonstrate that simulated microgravity, alone or in the presence of TLR7/8 agonist, can functionally alter cytokine production across immune cells. In general, consistent with sequencing data, the features demonstrate monocyte inflammatory function coupled to impaired T cell functionality in simulated microgravity. Thus, changing in cytokine signaling observed in simulated microgravity may occur at least in part to changes in upstream cytokine production.

Finally, to better assess individual immune cell sensitivity to stimulation, we have also reanalyzed our single cell data and added in a large volume of new data as new Supplemental Figures 12 and 13. This new analysis reinforces the idea of impaired interferon functional responses, especially in T cells and NK cells to simulated microgravity, further validating our Luminex, ELISA, flow cytometry, and existing sequencing data. See our response to Reviewer 1, general comment 1 for details.

4) It is unclear what conclusions the authors draw from the trajectory data presented in Figure 1E. For trajectory inference, we used Monocle3's reversed graph embedding (RGE) method to construct a minimum spanning tree that represents the most likely developmental trajectory among our cells. This trajectory, or pseudotime, represents the developmental progression of cells, which allows us to identify the order and direction of cellular differentiation events. The nodes often represent critical junctures in the trajectory where cells decide to differentiate into one fate or another. Each branch, therefore, corresponds to a different cell fate or lineage.

Our observations indicate that simulated microgravity (uG) has a significant impact on the behavior of innate immune cells, particularly monocytes, as reflected by the diversity of trajectories within this cluster. This implies that these cells are undergoing a broad array of transcriptional responses, potentially as an adaptive mechanism to the uG environment. The multiple nodes and branches observed in the trajectory analysis suggest that these cells are engaging in complex differentiation processes, possibly transitioning into various subtypes or other cell types. Each node and branch may represent unique maturation stages or responses to diverse signals, signifying varied differentiation paths. For instance, a monocyte might transition through several stages before it differentiates into a macrophage or a dendritic cell, each stage potentially represented by a separate node. Thus, the intricate trajectories under uG conditions could reflect not only intermediary stages or cell types in monocyte differentiation but also the potential for a more robust or nuanced immune response, such as we demonstrated by the pathway analysis in Figure 1F that the monocytes exhibit more alterations in molecular signaling pathways. We have updated the main body text to better explain these results.

5) The meta transcriptional data on virus reactivation in Figure 1I is very superficially described. How was this done and what conclusions can be drawn? Can a similar signal be seen in the confirmatory data sets available?

Thank you for the comments. The meta-transcriptional reads count that represents virus reactivation in Figure 1I was obtained by using MTD pipeline, which was published in 2022¹⁶. Specifically, the viral and microbial reads counting method in MTD is primarily built upon the kraken2 taxonomic sequence classification method¹⁷, which uses k-mers based reads classification algorithm. It breaks the sequence into overlapping k-mers and searches for them in the reference database. Then it uses the lowest common ancestor (LCA) algorithm to assign the taxonomic id to the query sequence. In the first step, any reads that were classified as belonging to the human (host) were filtered out, and then the remaining reads (non-host) were further classified as viral or microbial species by the LCA. The reads counts were then combined with the host reads and analyzed in R with Seurat package and other customized scripts (in the method section and on the GitHub repository https://github.com/FEI38750/Immune_Dysfunction_in_Microgravity). The relative abundance (frequency) of a virus or microbe was determined by dividing its reads count by the total reads count (host and non-host) in that sample.

In this revised version of the manuscript, we validated the classification results by using a different tool called Magic-BLAST¹⁸, which is an alignment-based method with a fundamentally distinct algorithm for assigning the reads. Nevertheless, we could still detect increases in gammaretrovirus and *mycobacterium canettii* transcripts seen with MTD pipeline in our single-cell RNAseq data sets/ This new data is added in as Supplementary Fig. 2B, C. We have also added this description of the new alignment method for validation in the method section. However, due to the significant difference in sequencing depth between single-cell and bulk, which single-cell has more than 18 times deeper depth than bulk (see the figure below), and the generally low proportion of viral transcripts in the sample, we were unable to detect viral reads in our bulk RNAseq data sets. Since detecting viral reactivation in our samples required deep sequencing only performed in our single-cell samples, we have tentatively

removed the line on viral reactivation from our abstract to de-emphasize that finding, though the two analyses of the single-cell data remain in the main body of the paper to promote further research into the topic.

Comparison of sequencing reads number between single-cell RNA-seq and bulk RNA-seq. Y-axis shows the total reads number after quality control. (A) The comparison of the mean number of reads was performed by parametric two-tailed T-test. The mean reads number of single-cell RNAseq is 430,604,142 and bulk RNAseq is 23,255,188. ****p ≤ 0.0001 (B) The number of reads in each sample. Sample names starting with "sc_" was used for single-cell sequencing, and the others were used for bulk RNA-seq.

6) The title of emphasizes “dysfunction” of the immune system in microgravity and spaceflight. Was dysfunction really shown, or is “changes” a better way to describe the changes?

Thank you for the suggestion. Incorporating the new data from this reviewer, as well as the others, we have built up evidence to support the idea the immune system shows abnormal changes in functional parameters in simulated microgravity. Some of these parameters are predicted from pathway analysis which includes spaceflight validation; some are shown by the differing in response to TLR agonist at the cytokine or ROS production level; some are shown by abnormal movements of the cytoskeleton in simulated microgravity. Overall the data points to a basally increased inflammatory potential of monocyte lineage cells in simulated microgravity, coupled with reduced responsiveness of T cells and NK cells at the gene expression, signaling pathways, and cytokine/secreted product levels. Similar findings can be predicted to occur from our pathway analysis data during spaceflight, or have been described in spaceflight (e.g. increased IL-1b and IL-6)^{19–21}. These changes to immune system would be consistent with dysfunction (e.g. heightened basal inflammatory changes, coupled to reduced adaptive and NK immune cell sensitivity to stimulation) in simulated microgravity. Because the new data supports the

use of the word, “dysfunction”, we prefer to keep it, though we are also open to other suggestions (e.g. changing the word “dysfunction” to “alteration”) at the discretion of the editor.

7) Overall, the manuscripts has a somewhat preliminary feel and the writing could be more direct and to the point.

Thank you very much for your valuable input; we have deleted parts of the manuscript, where we sound too speculative. One example is removing the comments on sirtuins, NAD, and ATP from the discussion.

Reviewer #3 (Remarks to the Author):

This is a novel study that applies single cell genomics to study the association of microgravity with immunological dysfunction. They characterize altered genes and pathways across immune cells under basal and stimulated states with a Toll like Receptor-7/8 agonist. At basal state, simulated microgravity altered the transcriptional landscape across immune cells, with monocyte subsets showing most pathway changes. Results from single cell analysis were validated by RNA sequencing and super-resolution microscopy. They provide evidences that microgravity has an impact on pathways essential for optimal immunity. Finally, they used machine learning to identify compounds can reverse abnormal pathways induced by microgravity. The paper reads interesting. It should be considered for publication. However, I have only one comment on the single cell experiment design.

“unstimulated PBMCs single-cell transcriptomes (10X Genomics), pooled together from a male (36 years old) and a female (25 years old) donor, that underwent either 1G or simulated microgravity (uG) for 25 hours total.” Lack of replicates is a big problem for the single cell experiment. However, the authors are very lucky that they pooled PBMC from one male and one female. It should be fairly easy to differentiate male single cells from female single cells using the RNA-seq raw data (by Y chromosome gene or SNP). In this way, the authors should be able add statistical power to their results. It should also be possible then to evaluate the batch effect and male female difference in their experiment. Lots of analyses need to be revised after separating male data from the female data.

Thank you for your feedback. We appreciate the suggestion to leverage sex-specific markers in our single-cell analysis to evaluate potential effects of sex differences. We would like to clarify first that our samples were not pooled for sequencing. Instead, the female and male samples were sequenced separately, with read counts conducted individually for each. The datasets were then integrated for subsequent analysis of overall gene expression alterations across both donors in response to microgravity.

In light of your comments, we have performed additional analyses to separate the data by sex, allowing us to study any potential differences in the responses to microgravity between the male and female samples. These findings can be found in a new Supplementary Figure 14 as well as in new Supplementary Table 10. Briefly, in the unstimulated state, female cells showed slightly more DEGs induced, while males had slightly more genes reduced. Male NK cells and monocytes were more sensitive to microgravity, while female B cells showed more sensitivity than male B cells. Upon

stimulation, male cells overall were more sensitive to simulated microgravity, especially in having more downregulated DEGs. The DEGs are also displayed, as well as a comparison of major pathways across immune cells. Overall, sex had mostly minor effects on pathway analysis, with most pathways altered similarly between male and female cells. For this reason, we have kept the analysis pooled in other figures. We also added in new bulk RNA sequencing data from 3 female donors, such that this validation cohort (now n=6) contains equal numbers of male and female donors so that both sexes are equally represented in the data.

However, as the Reviewer points out, there is a caveat, that the factor of sex is collinear with the batch, as the first single-cell sequencing batch comprises solely the female sample and the second batch consists of the male sample only. Consequently, we were unable to fully get rid of the batch effect, apart from the sex effect.

We hope that this clarification and additional analysis should address your concerns, and we appreciate your critical insight which has helped strengthen our study.

References

1. Harrington, C. A. *et al.* RNA-Seq of human whole blood: Evaluation of globin RNA depletion on Ribo-Zero library method. *Sci. Rep.* **10**, 6271 (2020).
2. Li, D. *et al.* Quercetin alleviates ferroptosis of pancreatic β cells in type 2 diabetes. *Nutrients* **12**, (2020).
3. Ortega, R. & García, N. The flavonoid quercetin induces changes in mitochondrial permeability by inhibiting adenine nucleotide translocase. *J. Bioenerg. Biomembr.* **41**, 41–47 (2009).
4. Jiang, J.-J., Zhang, G.-F., Zheng, J.-Y., Sun, J.-H. & Ding, S.-B. Targeting Mitochondrial ROS-Mediated Ferroptosis by Quercetin Alleviates High-Fat Diet-Induced Hepatic Lipotoxicity. *Front. Pharmacol.* **13**, 876550 (2022).
5. Bouamama, S. & Bouamama, A. Quercetin handles cellular oxidant/antioxidant systems and mitigates immunosenescence hallmarks in human PBMCs: An in vitro study. *J. Biochem. Mol. Toxicol.* e23354 (2023) doi:10.1002/jbt.23354.
6. Peng, J. *et al.* Quercetin Reprograms Immunometabolism of Macrophages via the SIRT1/PGC-1 α Signaling Pathway to Ameliorate Lipopolysaccharide-Induced Oxidative Damage. *Int. J. Mol. Sci.* **24**, (2023).
7. Li, Y. *et al.* Quercetin, inflammation and immunity. *Nutrients* **8**, 167 (2016).
8. Wang, W.-W. *et al.* Administration of quercetin improves mitochondria quality control and protects the neurons in 6-OHDA-lesioned Parkinson's disease models. *Aging (Albany NY)* **13**, 11738–11751 (2021).
9. Waseem, M. *et al.* Modulatory Role of Quercetin in Mitochondrial Dysfunction in Titanium Dioxide Nanoparticle-Induced Hepatotoxicity. *ACS Omega* **7**, 3192–3202 (2022).

10. Rayamajhi, N. *et al.* Quercetin induces mitochondrial biogenesis through activation of HO-1 in HepG2 cells. *Oxid. Med. Cell. Longev.* **2013**, 154279 (2013).
11. Du, H. *et al.* Tuning immunity through tissue mechanotransduction. *Nat. Rev. Immunol.* **23**, 174–188 (2023).
12. Uzer, G., Rubin, C. T. & Rubin, J. Cell mechanosensitivity is enabled by the LINC nuclear complex. *Curr. Mol. Biol. Rep.* **2**, 36–47 (2016).
13. Touchstone, H. *et al.* Recovery of stem cell proliferation by low intensity vibration under simulated microgravity requires LINC complex. *NPJ Microgravity* **5**, 11 (2019).
14. An, R. MRTF may be the missing link in a multiscale mechanobiology approach toward macrophage dysfunction in space. *Front. Cell Dev. Biol.* **10**, 997365 (2022).
15. Spatz, J. M. *et al.* Human immune system adaptations to simulated microgravity revealed by single-cell mass cytometry. *Sci. Rep.* **11**, 11872 (2021).
16. Wu, F., Liu, Y.-Z. & Ling, B. MTD: a unique pipeline for host and meta-transcriptome joint and integrative analyses of RNA-seq data. *Brief. Bioinformatics* **23**, (2022).
17. Wood, D. E., Lu, J. & Langmead, B. Improved metagenomic analysis with Kraken 2. *Genome Biol.* **20**, 257 (2019).
18. Boratyn, G. M., Thierry-Mieg, J., Thierry-Mieg, D., Busby, B. & Madden, T. L. Magic-BLAST, an accurate RNA-seq aligner for long and short reads. *BMC Bioinformatics* **20**, 405 (2019).
19. Gertz, M. L. *et al.* Multi-omic, Single-Cell, and Biochemical Profiles of Astronauts Guide Pharmacological Strategies for Returning to Gravity. *Cell Rep.* **33**, 108429 (2020).

20. Crucian, B. *et al.* Alterations in adaptive immunity persist during long-duration spaceflight. *NPJ Microgravity* **1**, 15013 (2015).
21. Slavish, D. C., Graham-Engeland, J. E., Smyth, J. M. & Engeland, C. G. Salivary markers of inflammation in response to acute stress. *Brain Behav. Immun.* **44**, 253–269 (2015).

REVIEWERS' COMMENTS

Reviewer #1 (Remarks to the Author):

I thank the authors for their careful attention regarding the previous round of comments. The manuscript is improved, and my outstanding questions have been addressed. The increased sample size and inclusion of analyses comparing responses across sex groups increased the rigor of the findings.

Reviewer #2 (Remarks to the Author):

The manuscript by Wu et al has been substantially strengthened by revisions and additional data.

However, I feel there is still room to improve the clarity of presentation. For example, in their rebuttal letter that authors clearly state an overall take home message "... the features demonstrate monocyte inflammatory function coupled to impaired T cell functionality in simulated microgravity". In the actual paper the reader has to dig for a while before this message becomes clear. Also some other take home messages are unnecessarily hidden.

Even though the text has been condensed a bit, there is still room for improvement in some parts that can be more concise and to the point.

The figures are a bit overloaded and it would help to divide the four figures into perhaps six.

On lines 1140-1141 it is stated that "...transcriptomic data from the Inspiration 4 mission was shared by Dr. Christopher Mason ...". This sounds a bit odd given that Dr Mason is an author on the manuscript.

Reviewer #3 (Remarks to the Author):

I have no further comments.

ADDITIONAL REVIEWER COMMENTS:

Reviewer #1 (Remarks to the Author):

I thank the authors for their careful attention regarding the previous round of comments. The manuscript is improved, and my outstanding questions have been addressed. The increased sample size and inclusion of analyses comparing responses across sex groups increased the rigor of the findings.

Thank you!

Reviewer #2 (Remarks to the Author):

The manuscript by Wu et al has been substantially strengthened by revisions and additional data.

However, I feel there is still room to improve the clarity of presentation. For example, in their rebuttal letter that authors clearly state an overall take home message "... the features demonstrate monocyte inflammatory function coupled to impaired T cell functionality in simulated microgravity". In the actual paper the reader has to dig for a while before this message becomes clear. Also some other take home messages are unnecessarily hidden.

Even though the text has been condensed a bit, there is still room for improvement in some parts that can be more concise and to the point.

The figures are a bit overloaded and it would help to divide the four figures into perhaps six.

On lines 1140-1141 it is stated that "...transcriptomic data from the Inspiration 4 mission was shared by Dr. Christopher Mason ...". This sounds a bit odd given that Dr Mason is an author on the manuscript.

Thank you for the suggestions.

To improve the clarity of the paper we removed a few more isolated speculative words like "might be", especially when cited literature supports the facts. An example is in the discussion, when talking about the relationship between reduced oxidative phosphorylation and increased glycolysis in immune cells, we changed "might be" to "can be". We have also condensed the text in scattered areas of the discussion. Importantly, to improve the clarity of the major take home message of "the features demonstrate monocyte inflammatory function coupled to

impaired T cell functionality in simulated microgravity" we have now added it right up front in the abstract so readers no longer have to search for it within the body of the paper.

At your advice, we have now split the original figure 1,2,3 into six figures, raising the total to 7 figures. We also reworded line 1140-1141 to state that 'data from the Inspiration 4 mission was collected by Dr. Christopher Mason'.

Reviewer #3 (Remarks to the Author):

I have no further comments.

Thank you!

References

1. Harrington, C. A. *et al.* RNA-Seq of human whole blood: Evaluation of globin RNA depletion on Ribo-Zero library method. *Sci. Rep.* **10**, 6271 (2020).
2. Li, D. *et al.* Quercetin alleviates ferroptosis of pancreatic β cells in type 2 diabetes. *Nutrients* **12**, (2020).
3. Ortega, R. & García, N. The flavonoid quercetin induces changes in mitochondrial permeability by inhibiting adenine nucleotide translocase. *J. Bioenerg. Biomembr.* **41**, 41–47 (2009).
4. Jiang, J.-J., Zhang, G.-F., Zheng, J.-Y., Sun, J.-H. & Ding, S.-B. Targeting Mitochondrial ROS-Mediated Ferroptosis by Quercetin Alleviates High-Fat Diet-Induced Hepatic Lipotoxicity. *Front. Pharmacol.* **13**, 876550 (2022).
5. Bouamama, S. & Bouamama, A. Quercetin handles cellular oxidant/antioxidant systems and mitigates immunosenescence hallmarks in human PBMCs: An in vitro study. *J. Biochem. Mol. Toxicol.* e23354 (2023) doi:10.1002/jbt.23354.
6. Peng, J. *et al.* Quercetin Reprograms Immunometabolism of Macrophages via the SIRT1/PGC-1 α Signaling Pathway to Ameliorate Lipopolysaccharide-Induced Oxidative Damage. *Int. J. Mol. Sci.* **24**, (2023).
7. Li, Y. *et al.* Quercetin, inflammation and immunity. *Nutrients* **8**, 167 (2016).
8. Wang, W.-W. *et al.* Administration of quercetin improves mitochondria quality control and protects the neurons in 6-OHDA-lesioned Parkinson's disease models. *Aging (Albany NY)* **13**, 11738–11751 (2021).
9. Waseem, M. *et al.* Modulatory Role of Quercetin in Mitochondrial Dysfunction in Titanium Dioxide Nanoparticle-Induced Hepatotoxicity. *ACS Omega* **7**, 3192–3202 (2022).

10. Rayamajhi, N. *et al.* Quercetin induces mitochondrial biogenesis through activation of HO-1 in HepG2 cells. *Oxid. Med. Cell. Longev.* **2013**, 154279 (2013).
11. Du, H. *et al.* Tuning immunity through tissue mechanotransduction. *Nat. Rev. Immunol.* **23**, 174–188 (2023).
12. Uzer, G., Rubin, C. T. & Rubin, J. Cell mechanosensitivity is enabled by the LINC nuclear complex. *Curr. Mol. Biol. Rep.* **2**, 36–47 (2016).
13. Touchstone, H. *et al.* Recovery of stem cell proliferation by low intensity vibration under simulated microgravity requires LINC complex. *NPJ Microgravity* **5**, 11 (2019).
14. An, R. MRTF may be the missing link in a multiscale mechanobiology approach toward macrophage dysfunction in space. *Front. Cell Dev. Biol.* **10**, 997365 (2022).
15. Spatz, J. M. *et al.* Human immune system adaptations to simulated microgravity revealed by single-cell mass cytometry. *Sci. Rep.* **11**, 11872 (2021).
16. Wu, F., Liu, Y.-Z. & Ling, B. MTD: a unique pipeline for host and meta-transcriptome joint and integrative analyses of RNA-seq data. *Brief. Bioinformatics* **23**, (2022).
17. Wood, D. E., Lu, J. & Langmead, B. Improved metagenomic analysis with Kraken 2. *Genome Biol.* **20**, 257 (2019).
18. Boratyn, G. M., Thierry-Mieg, J., Thierry-Mieg, D., Busby, B. & Madden, T. L. Magic-BLAST, an accurate RNA-seq aligner for long and short reads. *BMC Bioinformatics* **20**, 405 (2019).
19. Gertz, M. L. *et al.* Multi-omic, Single-Cell, and Biochemical Profiles of Astronauts Guide Pharmacological Strategies for Returning to Gravity. *Cell Rep.* **33**, 108429 (2020).

20. Crucian, B. *et al.* Alterations in adaptive immunity persist during long-duration spaceflight. *NPJ Microgravity* **1**, 15013 (2015).
21. Slavish, D. C., Graham-Engeland, J. E., Smyth, J. M. & Engeland, C. G. Salivary markers of inflammation in response to acute stress. *Brain Behav. Immun.* **44**, 253–269 (2015).